# Quantitative assessment of fire and vegetation properties in simulations with fire-enabled vegetation models from the Fire Model Intercomparison Project

Stijn Hantson[1,2], Douglas I. Kelley[3], Almut Arneth[1], Sandy P. Harrison[4], Sally Archibald[5], Dominique Bachelet[6], Matthew Forrest[7], Thomas Hickler[7,8], Gitta Lasslop[7], Fang Li[9], Stephane Mangeon[10,a], Joe R. Melton[11], Lars Nieradzik[12], Sam S. Rabin[1], I. Colin Prentice[13], Tim Sheehan[6], Stephen Sitch[14], Lina Teckentrup[15,16], Apostolos Voulgarakis[10], Chao Yue[17].

[1] Karlsruhe Institute of Technology, Institute of Meteorology and Climate research, Atmospheric Environmental Research, Garmisch-Partenkirchen, Germany.

[2] Geospatial Data Solutions Center, University of California Irvine, CA 92697 Irvine, USA.

[3] UK Centre for Ecology & Hydrology, Wallingford OX10 8BB, UK.

[4] School of Archaeology, Geography and Environmental Sciences, University of Reading, Reading, UK.

[5] Centre for African Ecology, School of Animal, Plant and Environmental Sciences, University of the Witwatersrand, Private Bag X3, WITS, Johannesburg, 2050, South Africa.

[6] Biological and Ecological Engineering, Oregon State University, Corvallis, OR 97331, USA.

[7] Senckenberg Biodiversity and Climate Research Institute (BiK-F), Senckenberganlage 25, 60325 Frankfurt am Main, Germany.

[8] Institute of Physical Geography, Goethe-University, Altenhöferallee 1, 60438 Frankfurt am Main, Germany.

[9] International Center for Climate and Environmental Sciences, Institute of Atmospheric Physics, Chinese Academy of Sciences, Beijing, China.

[10] Department of Physics, Imperial College London, London, UK.

[11] Climate Research Division, Environment and Climate Change Canada, Victoria, BC V8W 2Y2, Canada

[12] Institute for Physical Geography and Ecosystem Sciences, Lund University, 22362 Lund, Sweden.

[13] AXA Chair of Biosphere and Climate Impacts, Grand Challenges in Ecosystem and the Environment, Department of Life Sciences and Grantham Institute – Climate Change and the Environment, Imperial College London, Silwood Park Campus, Buckhurst Road, Ascot SL5 7PY, UK.

[14] College of Life and Environmental Sciences, University of Exeter, Exeter EX4 4RJ, UK.

[15] ARC Centre of Excellence for Climate Extremes, University of New South Wales, Sydney, NSW, Australia.

[16] Climate Change Research Center, University of New South Wales, Sydney, NSW 2052, Australia.

[17] Laboratoire des Sciences du Climat et de l'Environnement, LSCE/IPSL, CEA-CNRS-UVSQ, Université Paris-Saclay, 91198 Gif-sur-Yvette, France.

[a] now at: Data 61, CSIRO, Brisbane, Australia


*Correspondence to*: Stijn Hantson (hantson.stijn@gmail.com)

**Abstract.** Global fire-vegetation models are widely used to assess impacts of environmental change on fire regimes and the carbon cycle, and to infer relationships between climate, land use, and fire. However, differences in model structure and parameterizations, in both the vegetation and fire components of these models, could influence overall model performance, and to date there has been limited evaluation of how well different models represent various aspects of fire regimes. The Fire Model Intercomparison Project (FireMIP) is coordinating the evaluation of state-of-the-art global fire models, in order to improve projections of fire characteristics and fire impacts on ecosystems and human societies in the context of global environmental change. Here we perform a systematic evaluation of historical simulations made by nine FireMIP models to quantify their ability to reproduce a range of fire and vegetation benchmarks. The FireMIP models simulate a wide range in global annual total burnt area (39-536 Mha), and global annual fire carbon emission (0.91-4.75 Pg C a$^{-1}$) for modern conditions (2002-2012), but most of the range in burnt area is within observational uncertainty (345-468 Mha). Benchmarking scores indicate that seven out of nine FireMIP models are able to represent the spatial pattern in burnt area. The models also reproduce the seasonality in burnt area reasonably well but struggle to simulate fire season length and are largely unable to represent inter-annual variations in burnt area. However, models that represent cropland fires see improved simulation of fire seasonality in the northern hemisphere. The three FireMIP models which explicitly simulate individual fires are able to reproduce the spatial pattern in number of fires, but fire sizes are too small in key regions and this results in an underestimation of burnt area. The correct representation of spatial and seasonal patterns in vegetation appears to correlate with a better representation of burnt area. The two older fire models included in the FireMIP ensemble (LPJ-GUESS-GlobFIRM, MC2) clearly perform less well globally than other models, but it is difficult to distinguish between the remaining ensemble members: some of these models are better at representing certain aspects of the fire regime, none clearly outperforms all other models across the full range of variables assessed.

## 1 Introduction

Fire is a crucial ecological process that affects vegetation structure, biodiversity, and biogeochemical cycles in all vegetated ecosystems (Bond et al., 2005; Bowman et al., 2016) and has serious impacts on air quality, health, and economy (e.g. Bowman et al., 2009; Lelieveld et al., 2015; Archibald et al., 2013). In addition to naturally occurring wildland fires, fire is also used as a tool for pasture management and to remove crop residues. Because fire affects a large range of processes within the Earth system, modules which simulate burnt area and fire emissions are increasingly included in dynamic global vegetation models (DGVMs) and Earth System Models (ESMs) (Hantson et al., 2016; Kloster and Lasslop, 2017; Lasslop et al., 2019). However, the representation of both lightning-ignited fires and anthropogenic fires (including cropland fires) varies greatly in global fire models. This arises due to the lack of a comprehensive understanding of how fire ignitions, spread, and suppression are affected by weather, vegetation, and human activities, as well as the relative scarcity of long-term, spatially resolved data on the drivers of fires and their interactions (Hantson et al., 2016). As a result, model projections of future fire are highly uncertain (Settele et al., 2014; Kloster and Lasslop, 2017). Since vegetation mortality – including fire-related death – is one determinant of carbon residence time in ecosystems (Allen et al., 2015), differences in the representation of fire in DGVMs or ESMs also contributes to the uncertainty in trajectories of future terrestrial carbon uptake (Ahlström et al., 2015; Friend et al., 2014; Arora & Melton, 2018). Improved projections of wildfires and anthropogenic fires, their impact on ecosystem properties, and their socio-economic

impact will therefore support a wide range of global environmental change assessments, as well as the development of strategies for sustainable management of terrestrial resources.

Although individual fire-enabled DGVMs have been evaluated against observations, comparisons of model performance under modern-day conditions tend to focus on a limited number of fire-related variables or specific regions (e.g. French et al., 2011; Wu et al., 2015; Ward et al., 2016; Kloster and Lasslop, 2017). Such comparisons do not provide a systematic evaluation of whether different parameterizations or levels of model complexity provide a better representation of global fire regimes than others. Likewise, none of the Coupled Model

Intercomparison Projects that have been initiated to support the IPCC process (CMIP: Taylor et al., 2012; Eyring et al., 2016) focuses on fire, even though several of the CMIP models simulate fire explicitly. The Fire Model Intercomparison Project (FireMIP) is a collaborative initiative to systematically evaluate state-of-the-art global fire models (Hantson et al., 2016; Rabin et al., 2017).

The FireMIP initiative draws on several different types of simulations, including a baseline historical simulation

(1700-2013 CE) and sensitivity experiments to isolate the response of fire regimes to individual drivers, as well as simulations in which fire is deliberately excluded (Rabin et al., 2017). While the sensitivity and exclusion experiments provide valuable insights into model behaviour (Teckentrup et al., 2019; Li et al., 2019), the baseline historical simulation provides an opportunity to assess how well the models simulate modern conditions. Model-model differences could reflect differences in the treatment of fire, of ecosystem processes, or how fire interacts

with other aspects of the land surface in an individual model. Evaluation of the baseline simulations needs therefore to include evaluation of ecosystem processes and diagnosis of interactions between simulated vegetation and fire.

Systematic model evaluation can also serve another purpose. The analysis of future climate and climate impacts is often based on results from climate and impact model ensembles (e.g. Kirtman et al., 2013; Collins et al., 2013; Warszawski et al. 2013) and these ensembles are also being used as a basis for impact assessments (e.g. Settele et

al., 2014; Hoegh-Guldberg et al., 2019). However, there is increasing dissatisfaction with the idea of using the average behaviour of model ensembles without accounting for the fact that some models are less reliable than others (Giorgi and Mearns 2002; Knutti, 2010; Parker et al., 2013) and many have called for "the end of model democracy" (e.g. Held, 2005; Knutti, 2010). Although there is still considerable discussion about how to constrain models using observations, and then how to combine and possibly weight models depending on their overall

performance or performance against a minimum set of specific criteria (e.g. Eyring et al., 2005; Tebaldi et al., 2005; Gleckler et al., 2008; Weigel et al., 2008; Santer et al., 2009; Parker, 2013; Abramowitz et al., 2019), it is clear that results from systematic evaluations are central to this process.

A number of papers have examined specific aspects of the FireMIP baseline simulations. Andela et al. (2017) showed that the FireMIP models do not reproduce the decrease in global burnt area over the past two decades

inferred from analysis of version 4s of the Global Fire Emission Database (GFED4s) data product. In fact, four of the models show an increase in burnt area over the period 1997-2014. Although the remaining five models show a decrease, their mean decrease is only about one tenth of the observed rate ($-0.13 \pm 0.56\%$ yr$^{-1}$, compared to the observed trend of $-1.09 \pm 0.61\%$ yr$^{-1}$). However, the observed global decline of burnt area derived from satellite data is strongly dominated by African savanna ecosystems, the spatial pattern of trends is very heterogeneous, and

the satellite record is still very short, which raises issues about the robustness of these trends (Forkel et al., 2019b). Li et al. (2019) compared modelled and satellite-based fire emissions and concluded that most FireMIP models fall within the current range of observational uncertainty. Forkel et al. (2019a) compared the emergent relationships between burnt area and multiple potential drivers of fire behaviour, including human caused ones, as seen in

observations and the FireMIP models. They show that, although all of the models capture the observed emergent relationships with climate variables, there are large differences in their ability to capture vegetation-related relationships. This is underpinned by a regional study using the FireMIP models over China, which showed that there are large differences in simulated vegetation biomass, and hence in fuel loads, between the models (Song et al., 2019). These results make a focus on benchmarking both simulated fire and vegetation particularly pertinent. Forkel et al. (2019a) showed that some of the FireMIP models, specifically those that include a relatively strong fire suppression associated with human activities (Teckentrup et al., 2019), were able to reproduce the emergent relationship with human population density. However, the treatment of the anthropogenic influence on burnt area has been identified as a weakness in the FireMIP models (Andela et al., 2017; Teckentrup et al., 2019; Li et al., 2019; Forkel et al., 2019a), mainly due to a lack of process understanding.

In this paper, we focus on quantitative evaluation of model performance using the baseline historical simulation and a range of vegetation and fire observational datasets. We use the vegetation-model evaluation framework described by Kelley et al. (2013), with an extended set of data targets to quantify the fire and vegetation properties and their uncertainties. We identify (i) common weaknesses of the current generation of global fire-vegetation models, (ii) factors causing differences between the models, and (iii) discuss the implications for future model development.

## 2 Methods

### 2.1 Model Simulations

The baseline FireMIP simulation is a transient experiment starting in 1700 CE and continuing to 2013 (see Rabin et al. (2017) for description of the modelling protocol and the sources of the input data for the experiments). Models were spun up until carbon stocks were in equilibrium for 1700 CE conditions (equilibrium was defined as <1% change over a 50 year time period for the slowest carbon pool in each grid cell) using land use and population density for 1700 CE, $CO_2$ concentration for 1750 CE, and recycling climate and lightning data from 1901-1920 CE. Although the experiment is fully transient after 1700 CE, annually varying values of all these forcings are not available until after 1900 CE. Climate, land use, population and lightning were regridded to the native grid of each model. Global fire-vegetation models ran with either dynamic or prescribed natural vegetation (Table 1), but all used observed time-evolving cropland and pasture (if simulated) distribution.

Nine coupled fire-vegetation models have performed the FireMIP baseline experiments. The models differ in complexity, representation of human impact and vegetation dynamics, and spatial and temporal resolution (Table 1). A detailed description of each model is given in Rabin et al. (2017). Most of the models ran simulations for the full period 1700-2013, but CLASS-CTEM, JULES-INFERNO, MC2 and CLM simulated 1861-2013, 1700-2012, 1902-2009 and 1850-2013 respectively. This slight deviation from the protocol does not affect the results all but 1 model presented here as we only analyse data for present-day period (2002-2012). For MC2 the 2002-2009 time period was used for analysis which might influence the results for this model.

### 2.2 Benchmarking reference datasets

Model performance was evaluated using site-based and remotely sensed global data sets of fire occurrence, fire-related emissions and vegetation properties (Figure 1; Figure S1). We include vegetation variables (e.g. GPP,

NPP, biomass, LAI) because previous analyses have indicated that they are critical for simulating fire occurrence and behaviour (Forkel et al., 2019a; Teckentrup et al., 2019) and there are global data sets available. We did not consider parameters such as soil or litter moisture because, although these may have an important influence on fire behaviour, globally comprehensive data sets are not available. All datasets are plotted in Figure S1. We used multiple datasets as targets for variables where they were available in order to take into account observational uncertainty.

Ideally, model benchmarking should take account of uncertainties in the observations. However, observational uncertainties are not reported for most of the data sets used here (e.g. vegetation carbon). While it would in principle be possible to include uncertainty for example by down-weighting less reliable data sets (e.g. Collier et al. 2018), determining the merits of the methods used to obtain observational data is rather subjective and no agreement as to which is more reliable if multiple reference datasets exist for the same variable (e.g. burnt area). Furthermore, some of the data sets (e.g. emissions) involve modelled relationships; there has been little assessment of the impact of the choice of model on the resultant uncertainty in emission estimates (e.g. Kaiser et al., 2012). While we use multiple datasets when available (e.g. for burnt area, where there are extremely large differences between the products and they may all underestimate the actual burnt area (Roteta et al., 2019)), in an attempt to integrate observational uncertainty in our evaluations, it seems premature to incorporate uncertainty in the benchmark data sets in a formal sense in calculating the benchmarking scores.

The following datasets where used for model evaluation:

Burnt Area:

Five global burnt fraction products were used in this study (Figure S1). We used the fourth version of the Global Fire Emissions Database (GFED4) for 1997-2013, which uses the MCD64 burnt area MODIS based product in combination with an empirical estimation of burnt area based on thermal anomalies when MODIS data was unavailable (Giglio et al., 2013). We also included a version where the MCD64 burnt area product was merged with the small fire detection approach developed by Randerson et al. (2012; GFED4s). The third dataset is the MODIS burnt area product MCD45 which is the only burnt area product not using MODIS thermal anomalies within its burnt area detection algorithm (2002-2013) (Roy et al., 2008). The fourth is the FireCCIv4.0 dataset based on MERIS satellite data (Alonso-Canas and Chuvieco, 2015), available for the period 2005-2011. The fifth is the FireCCI5.1 dataset based on MODIS 250m imagery (Chuvieco et al., 2018).

Fire emissions:

Carbon emission by fires is estimated within the Global Fire Assimilation System (GFAS) based on satellite-retrieved fire radiative power (FRP) (Kaiser et al., 2012). Here we use the global GFAS data for the period 2000-2013.

Fire size and numbers:

Estimates on mean size and number of fires can be produced using a floodfilling algorithm to extract individual fires (Archibald et al., 2013). Here we use the data as produced by Hantson et al. (2015) from the MCD45 global burnt area product (Roy et al., 2008). Only large fires ≥25ha (one MODIS pixel) are detected, with a considerable underestimation of fires < ~125ha. Therefore, a direct comparison with modelled fire numbers and size is meaningless, but evaluation of the spatial pattern in fire numbers and fire size can be performed.

Vegetation productivity:

We use multiple datasets for vegetation productivity, both measurements from site locations and global upscaled estimates. The site-level GPP dataset is from Luyssaert et al (2007) and the site-level NPP combines these

data with data from the Ecosystem Model/Data Intercomparison (EMDI; (Olson et al., 2001)) databases. Sites from managed or disturbed environments were not used. A recent compilation of NPP site-level estimates was compiled by Michaletz et al. (2014). The mean of observations was taken when more than 1 measurement was available within a 0.5º grid cell. We also use upscaled fluxnet GPP data (Jung et al., 2017; Tramontana et al., 2017). Kelley et al. (2013) showed that the spreading of data between fluxnet site observations in such upscaling artificially improved model performance, probably because it used similar input data and using methods which might emulate functional relationships used within DGVMs. Hence scores obtained by Jung should not be interpreted as true "benchmarking scores" but could help inform differences between models in relation to scores obtained from other comparisons like burnt area (See Figure S1).

Carbon in vegetation:

A global dataset on aboveground vegetation biomass was recently produced by combining two existing datasets—Saatchi et al. (2011) and Bacchini et al. (2012)—using a reference dataset of field observations and estimates (Avitabile et al., 2016). However, this dataset only considers woody biomass and to be able to analyse vegetation carbon also for areas without tree cover we used the dataset generated by Carvalhais et al. (2003) whom combined the Saatchi et al. (2011) and Thurner et al. (2004) biomass datasets while providing a best estimate for herbaceous biomass.

Leaf Area Index (LAI):

We use the MODIS LAI product MCD15 which gives global LAI values each 8 days (Myneni et al., 2002) and the LAI dataset produced based on AVHRR (Claverie et al., 2016). The mean LAI over the period 2001-2013 is used for benchmarking.

**2.3) Model Evaluation and Benchmarking**

We adopted the metrics and comparison approach specified by Kelley et al. (2013) as it provides a comprehensive scheme for the evaluation of vegetation models. This protocol provides specifically designed metrics to quantify model performance in terms of annual average, seasonal and inter-annual variability against a range of global data sets, allowing the impact of spatial and temporal biases in means and variability to be assessed separately. The derived model scores were compared to scores based on the temporal or spatial mean value of the observations and a "random" model produced by bootstrap resampling of the observations.

NME was selected over other metrics (e.g. RMSE) as these normalised scores allow for direct comparison in performance between variables with different units (Kelley et al., 2013), NME is more appropriate for variables which do not follow a normal distribution and it has therefore been used as the standard metric to asses global fire model performance (e.g, Kloster & Lasslop, 2017; kelley et al., 2019; Boer et al., 2020) . NME is defined as:

$$NME = \frac{\sum A_i |obs_i - sim_i|}{\sum A_i |obs_i - \overline{obs}|} \tag{1}$$

where the difference between observations (obs) and simulation (sim) are summed over all cells (i) weighted by cell area (Ai) and normalized by the average distance from the mean of the observations ($\overline{obs}$). Since NME is proportional to mean absolute errors, the smaller the NME value the better the model performance. A score of 0 represents a perfect match to observations. NME has no upper bound.

NME comparisons were conducted in three steps following Kelley et al. (2013):

Step 1. As described above;

Step 2. With obs$_i$ and sim$_i$ replaced with the difference between observation or simulation and their respective means. ie $x_i \rightarrow x_i - \bar{x}$ , removing systematic bias and describe the performance of the model around the mean.

Step 3. Where obs$_i$ and sim$_i$ from step 2 were divided by the mean deviation. i.e $x_i \rightarrow x_i/|x_i|$. This removed the influence of bias in the variability and described the models ability to reproduce the spatial pattern in burnt area.

To limit the impact of observational uncertainties in the reference datasets on the comparisons and as NME can be sensitive to the simulated magnitude of the variable we mainly focus on benchmarking results after removing the influence of biases in the mean and variance (step 3). However, comparisons step 1 and 2 are given in Supplementary Information Table S1.

To assess a model's ability to reproduce seasonal patterns in a variable, we focused on seasonal concentration (roughly equivalent to the inverse of season length) and seasonal phase (or timing) comparisons from Kelley et al. (2013). This uses the mean seasonal "vector" for each observed and simulated location based on the monthly distribution of the variable through the year. whereby each month, m, is represented by a vector in the complex plane whose direction ($\theta_m$) corresponds to the time of year and length to the magnitude of the variable for that month:

$$\theta_m = 2 \cdot \pi \cdot (m - 1)/12 \qquad (2)$$

A mean vector L is the average of the real (L$_x$) and imaginary (L$_y$) parts of the 12 vectors (x$_m$).

$$L_x = \Sigma_m x_m \cdot \cos(\theta_m) \qquad (3)$$
$$L_y = \Sigma_m x_m \cdot \sin(\theta_m)$$

The mean vector length by the annual average describes the seasonal concentration (C) of the variable, while it's direction (P) describes seasonal timing (phase):

$$C = \frac{\sqrt{L_x^2 + L_y^2}}{\Sigma_m x_m} \qquad (4)$$

$$P = \arctan\left(\frac{L_x}{L_y}\right) \qquad (5)$$

If the variable in a given cell is concentrated all in one month, then, C is equal to 1 and P corresponds to that month. If burnt area is evenly spread throughout the year then concentration is zero and phase is undefined. Where the phase of a cell is undefined in either observations or simulation, then it was not used in the comparison. Likewise, if a cell has zero annual average burnt area for either observations or simulation, then that cell is not included in the comparisons. Concentration was compared using NME step 1. Phase was compared using the mean phase difference metric (MPD):

$$MPD = \frac{1}{\pi}\Sigma_i A_i \cdot arcos\left[cos\left(P_{sim,i} - P_{obs,i}\right)\right]/\Sigma_i A_i \qquad (6)$$

MPD represents the average timing error, as a proportion of the maximum phase mismatch (6 months).

Seasonality metrics could not be calculated for three models (LPJ-GUESS-GlobFIRM, LPJ-GUESS-SIMFIRE-BLAZE, MC2), either because they do not simulate the seasonal cycle or because they did not provide these

outputs. We did not use FireCC4.0 to assess seasonality or interannual variability (IAV) in burnt area because it has a much shorter times series than the other burnt area products.

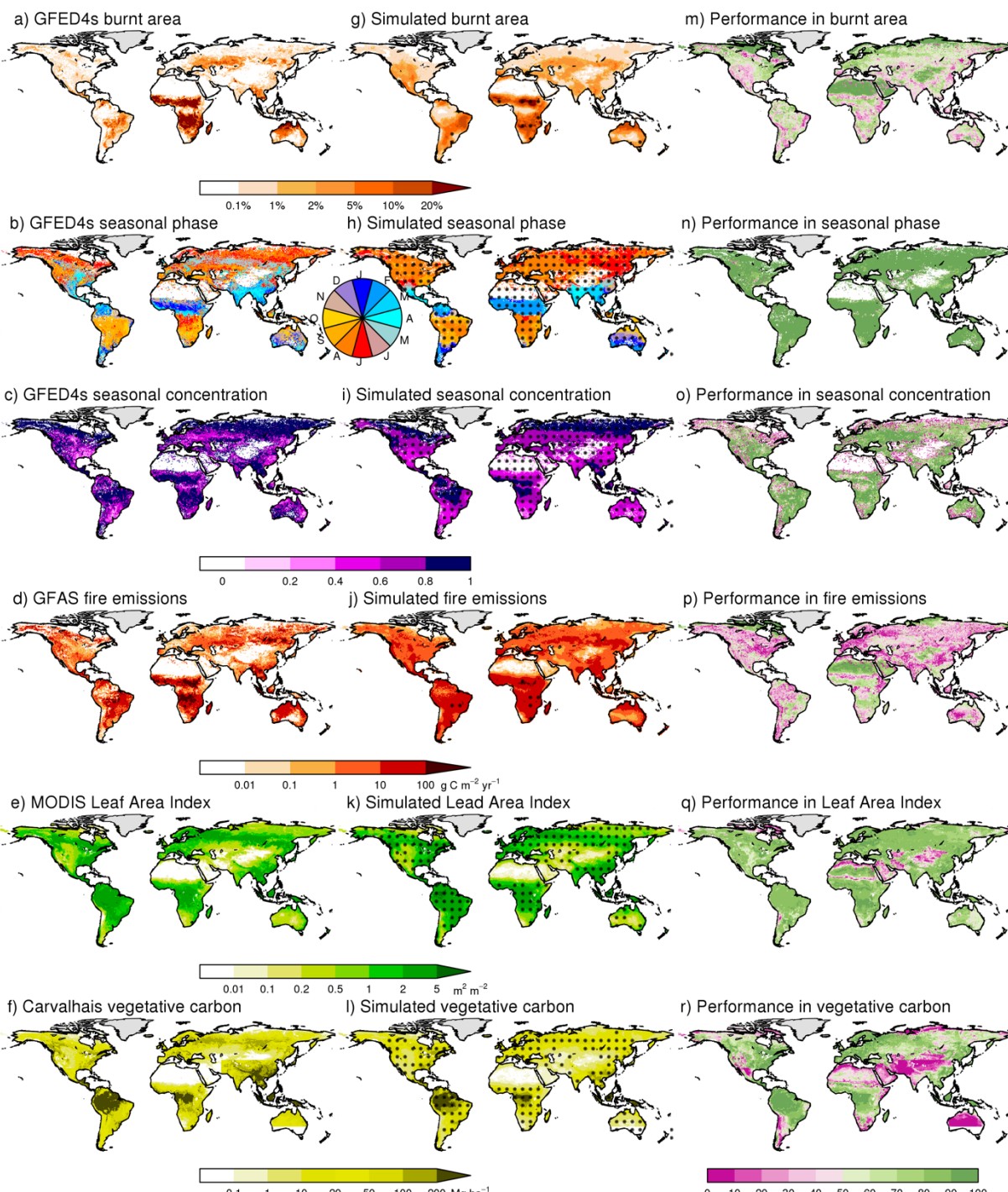

**Figure 1: Reference datasets, the mean of all models, and the % of models for which the estimate falls within 50-200% of the (mean) reference data are presented for a set of fire relevant variables. Results for the following variables are given: a) fraction burnt area; b) seasonal timing of burnt area (as measured by mean phase); c) burnt area season length (as measured by seasonal concentration); d) fire C emissions (g C m$^{-2}$ yr$^{-1}$); e) vegetation carbon (Mg/ha); and f) Leaf Area Index (LAI) (m$^2$/m$^2$). Stippling in the 2nd column indicates where variance between models is less than the FireMIP**
**model ensemble mean. Purple in the 3rd column indicates cell where the majority of the FireMIP models produce poor simulations of the variable, while green areas indicate that the majority of the FireMIP models perform well for that aspect of the fire regime.**

Model scores are interpreted by comparing them to two null models (Kelley et al., 2013). The "mean" null model compares each benchmark dataset to a dataset of the same size created using the mean value of all the observations. The mean null model for NME always has a value of 1 because the metric is normalised by the mean difference. The mean null model for MPD is based on the mean direction across all observations, and therefore the value can vary and is always less than 1. The "randomly-resampled" null model compares the benchmark data set to these observations resampled 1000 times without replacement (Table 3). The "randomly-resampled" null model is normally worse than the mean null model for NME comparisons. For MPD, the mean will be better than the random null model when most grid cells show the same phase. A detailed description of the benchmarking metrics is given in the Supplementary Information S2.

For comparison and application of the benchmark metrics, all the target datasets and model outputs were resampled to a 0.5° grid. Although some models were run at a coarser resolution, the spatial resolution at which the benchmarking was performed had only a limited impact on the scores (Figure S2), which does not affect conclusions drawn here. Each model was compared to each reference dataset except in the few cases where the appropriate model output was not provided (e.g. LAI in ORCHIDEE, GPP in MC2). Only the models which incorporate the SPITFIRE fire module provided fire size and number results.

## 3 Results

### 3.1 Modern day model performance: burnt area and fire emissions

The simulated modern (2002-2012) total global annual burnt area is between 39 and 536 Mha (Table 2). Most of the FireMIP models are within the range of burnt area estimated by the individual remotely sensed products (354 to 468 Mha a–1). LPJ-GUESS-GlobFIRM and MC2 simulate much less burnt area than the shown by any of the products and CLASS-CTEM simulates more than shown by any of the products. However, use of the range of the remotely sensed estimates may not be a sufficient measure of the uncertainty in burnt area because four of them are derived from the same active fire product (Forkel et al., 2019) and recent work suggests that they may all underestimate burnt area (Roteta et al., 2019). Thus, we cannot definitively say that the apparent overestimation by CLASS-CTEM is unrealistic. With the exception of MC2 and LPJ-GUESS-GlobFIRM, the models realistically capture the spatial patterns in burnt area (Figures 1 & 2) and perform better than either of the null models irrespective of the reference burnt area dataset (Table 3). CLM (NME: 0.63-0.80) and ORCHIDEE-SPITFIRE (0.70-0.73) are the best performing models. All the FireMIP models correctly simulate most burnt area in the tropics (24-466 Mha a$^{-1}$) compared to observed values in the range 312-426 Mha a$^{-1}$ (Table 2). The simulated contribution of tropical fires to global burnt area is in the range of 56% to 92%, with all models except ORCHIDEE-SPITFIRE simulating a lower fraction than observed (89-93%). This follows from FireMIP models tending to underestimate burnt area in Africa and Australia, although burnt area in South American savannas is usually overestimated (Table 2). All of the FireMIP models, except LPJ-GUESS-GlobFIRM, capture a belt of high burnt area in central Eurasia. However, the models overestimate burnt area across the extratropics on average by 180% to 304%, depending on the reference burnt area dataset. This overestimation largely reflects the fact that the simulated burnt area over the Mediterranean basin and western USA is too large (Table 2, Figure 2).

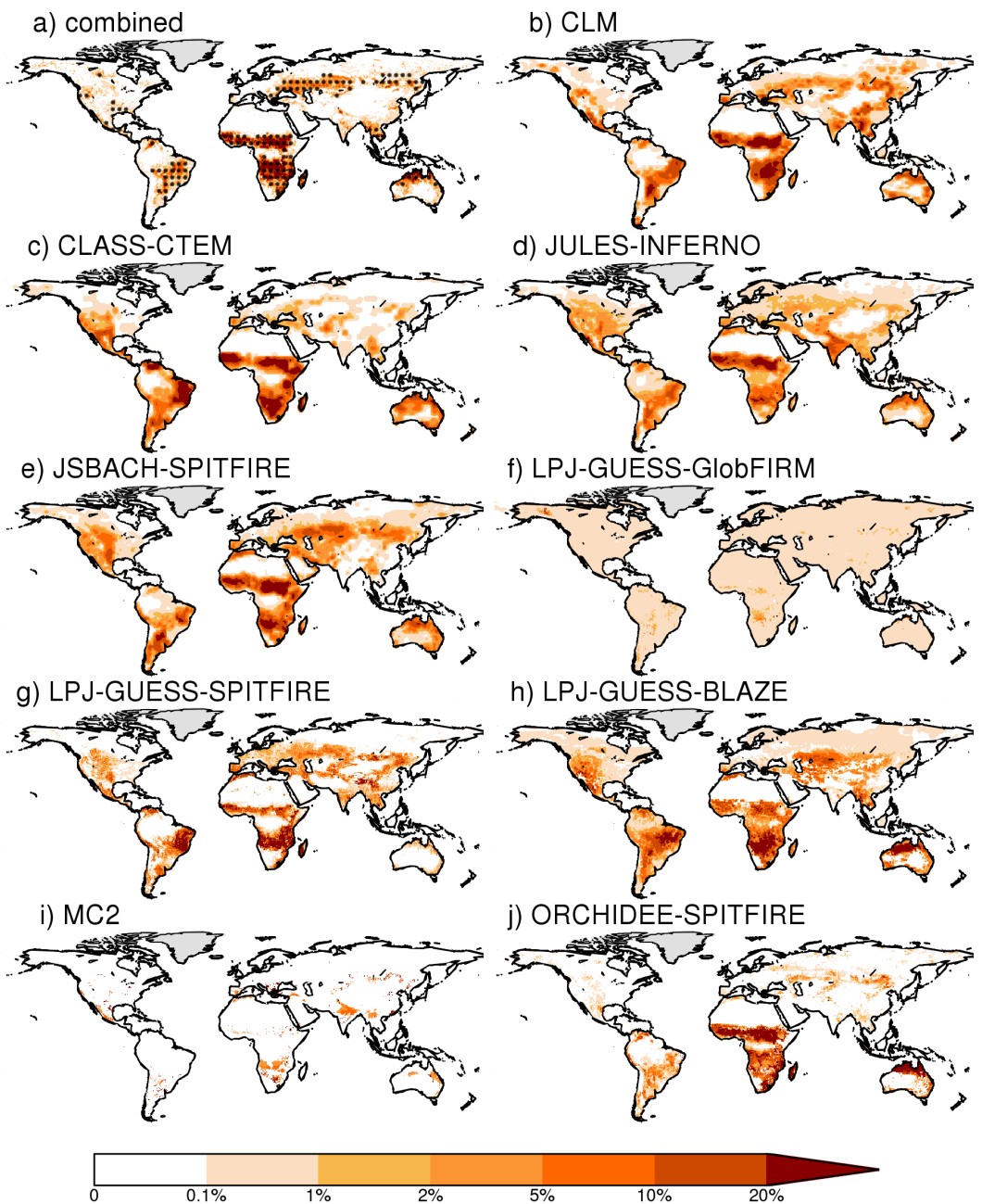

**Figure 2: Simulated versus observed burnt fraction (% yr⁻¹) for the present day (2002-2012), where "combined" indicates the mean of the different burnt area datasets considered. Stippling indicates where variance between burnt area datasets is less than the observed ensemble mean.**

The FireMIP models that include a sub-annual time-step for fire calculations (CLM, CLASS-CTEM, JULES-INFERNO, JSBACH-SPITFIRE, LPJ-GUESS-SPITFIRE, ORCHIDEE-SPITFIRE) generally reproduce the seasonality of burnt area (Figure 3), particularly in the tropics. The models capture the timing of the peak fire season reasonably well, with all of the models performing better than mean null model for seasonal phase in burnt area (Table 3). The models also frequently perform better than the random null model, with all models performing better against GFED4. However, all of the FireMIP models perform worse than mean null model for seasonal concentration of burnt area, independent of the reference burnt area dataset. The observations show a unimodal pattern in burnt area in the tropics, peaking between November through February in the northern tropics and between June through October in the southern tropics (Figure 3). The models also show a unimodal pattern in both

regions. However, all the FireMIP models except ORCHIDEE-SPITFIRE show a ~2-month delay in peak burnt area in the northern tropics, and the period with high burnt area is also less concentrated than observed. Some models (ORCHIDEE-SPITFIRE, LPJ-GUESS-SPITFIRE) estimate peak burnt area ~1-2 months too early in the southern tropics, while others simulate a peak ~1 month too late (JULES-INFERNO, CLM, CLASS-CTEM) or have a less concentrated peak (JSBACH-SPITFIRE, JULES-INFERNO) than observed. The seasonality of burnt area in the northern extratropics shows a peak in spring and a second peak in summer. Only CLM reproduces this double peak, while all of the other FireMIP models show a single summer peak. Most of the models simulate the timing of the summer peak well. The only exception is LPJ-GUESS-SPITFIRE, which simulates the peak ~2-3 months too late. The observations show no clear seasonal pattern in burnt area over the southern extratropics, although the most prominent peak occurs in December and January. All the FireMIP models, except LPJ-GUESS-SPITFIRE, reproduce this mid-summer peak. LPJ-GUESS-SPITFIRE shows little seasonality in burnt area in this region.

The FireMIP models have problems representing IAV in global burnt area, with some models (CLASS-CTEM, MC2) worse than the random model and most models performing worse than the mean for most of the target data sets (Table 3). However, there is considerable uncertainty in the observed IAV in burnt area (Figure 4), and the scores are therefore dependent on the reference dataset considered, with generally worse scores for FireCCI5.1 and GFED4s compared to the other datasets. Observational uncertainty is most probably underestimated as the burnt area products are not independent, since they all rely on MODIS satellite imagery. Despite the failure to reproduce IAV in general, most of the models show higher burnt area in the early 2000s and a low in 2009-2010 after which burnt area increased again (Figure 4).

The spatial patterns in simulated fire-related carbon emissions are in line with the reference data, with most FireMIP models except LPJ-GUESS-GlobFIRM, MC2 and LPJ-GUESS-SPITFIRE performing better than the mean null model. CLM, JULES-INFERNO and JSBACH-SPITFIRE are the best performing models with NME scores < 0.8. Seasonality in fire emissions mimics the results for burnt area with good scores for seasonal phase, but all models perform worse than the mean null model for seasonal concentration. CLM is the only FireMIP model to explicitly include peatland, cropland and deforestation fires, which contribute 3%, 3% and 20% respectively of the global total emissions annually (van der Werf et al., 2010), but it nevertheless does not perform better than JULES-INFERNO and JSBACH-SPITFIRE in representing the spatial pattern of fire carbon emissions.

Only three models (JSBACH-SPITFIRE, LPJ-GUESS-SPITFIRE, ORCHIDEE-SPITFIRE) provided information about simulated numbers and size of individual fires. All three models performed better than the mean null model in representing the spatial pattern in number of fires but worse than the mean model for fire size (Table 3). While the spatial pattern in simulated fire number is in agreement with observations over large parts of the globe, models tend to overestimate fire numbers in dryland areas such as Mexico and the Mediterranean basin (Figure 5). None of the three models simulate cropland fires and so they do not capture the high number of cropland fires (Hall et al., 2016) in central Eurasia (Table 2). Models simulate smaller fires than observed in areas where burnt area is large and where models tend to underestimate burnt area, especially in the African savanna regions (Figure 5).

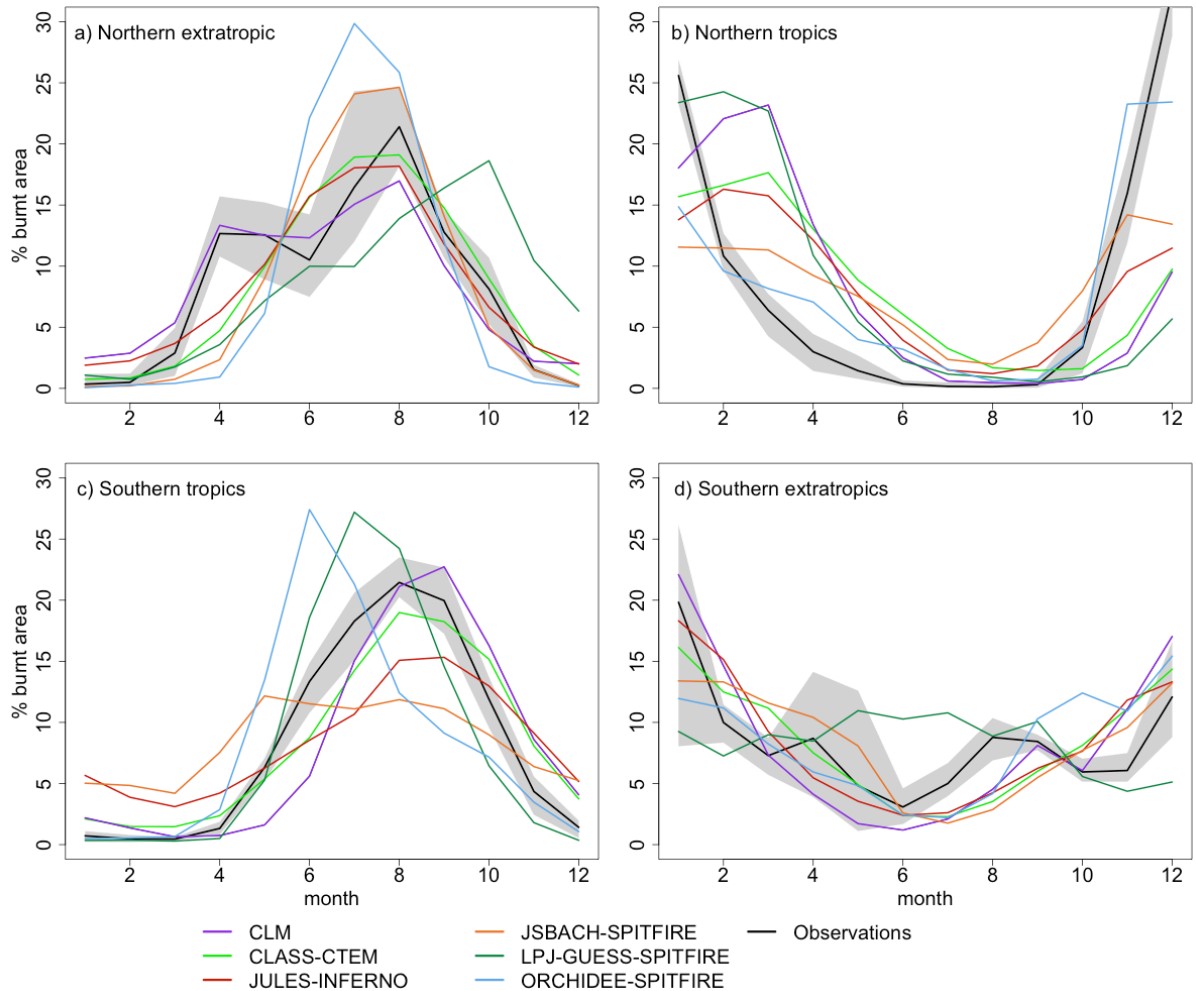

**Figure 3: Simulated and observed seasonality (2002-2012) of burnt area (% of annual burnt area per month) for a) northern extratropics (> 30°N), b) northern tropics (0-30°N), c) southern tropics (0-30°S) and d) southern extratropics (> 30°S). The mean of all the remotely sensed burnt area datasets is shown as a black line, with the minimum and maximum range shown in light grey.**

### 3.2. Present day model performance: Vegetation properties

Fire spread and hence burnt area is strongly influenced by fuel availability, which in turn is affected by vegetation primary production and biomass. Simulated spatial patterns of GPP compare well with estimates of GPP upscaled from Fluxnet data (Jung et al., 2017), with scores (0.39-0.67) considerably better than both null models. However, performance against site-based estimates of GPP (Luyssaert et al., 2007) are considerably poorer (1.09-1.49) and worse than the mean null model. Only LPJ-GUESS-SPITFIRE, LPJ-GUESS-SIMFIRE-BLAZE and ORCHIDEE-SPITFIRE perform better than the random null model. There is no clear relationship between model scores for the two datasets: models performing better when compared to the Jung dataset do not necessarily show a higher score when compared to the Luyssaert GPP dataset. The two GPP datasets are very different: The upscaled FLUXNET dataset is a modelled product but has global coverage (see Supplementary Information S1) while the Luyssaert dataset has local measurements but only at a limited number of sites, largely concentrated across the northern extratropics. Thus, the better match between the FireMIP models and the upscaled FLUXNET dataset may reflect the broader spatial coverage or the fact that climate and landcover data are used for upscaling.

Only the upscaled Fluxnet data provides monthly data and can thus be used to asses GPP seasonality. The FireMIP models are able to represent the seasonal peak timing in GPP, with all models performing better than the

mean and random null models. However, models have difficulty in representing the length of the growing season, with the scores for seasonal concentration in GPP (1.08-1.23) above the mean null model but below the random null model for all FireMIP models.

Model performance is better for site-level NPP than site-level GPP. All of the FireMIP models perform better than the mean null model, independent of the choice of reference data set (Table 3), except for CLASS-CTEM against the Luyssaert data set. JULES-INFERNO, JSBACH-SPITFIRE and MC2 are the best-performing models.

The FireMIP models generally capture the spatial pattern in LAI, with all models performing better than the mean null model (0.44-0.81), independent of the reference dataset considered. JULES-INFERNO has the best

score for both reference datasets. Although the overall global pattern in LAI is well represented in all the FireMIP models, they have more trouble representing LAI in agricultural areas such as central USA or areas with low LAI such as drylands and mountain areas (Figure 1).

The FireMIP models perform well in representing the spatial pattern carbon in vegetation (Table 3). All nine models perform better than the mean null model, independent of reference dataset, with ORCHIDEE-SPITFIRE

having the best scores. Generally, the models are able to simulate carbon in tropical vegetation and the forested regions in the temperate and boreal region reasonably well, but struggle across most dryland systems (Figure 1).

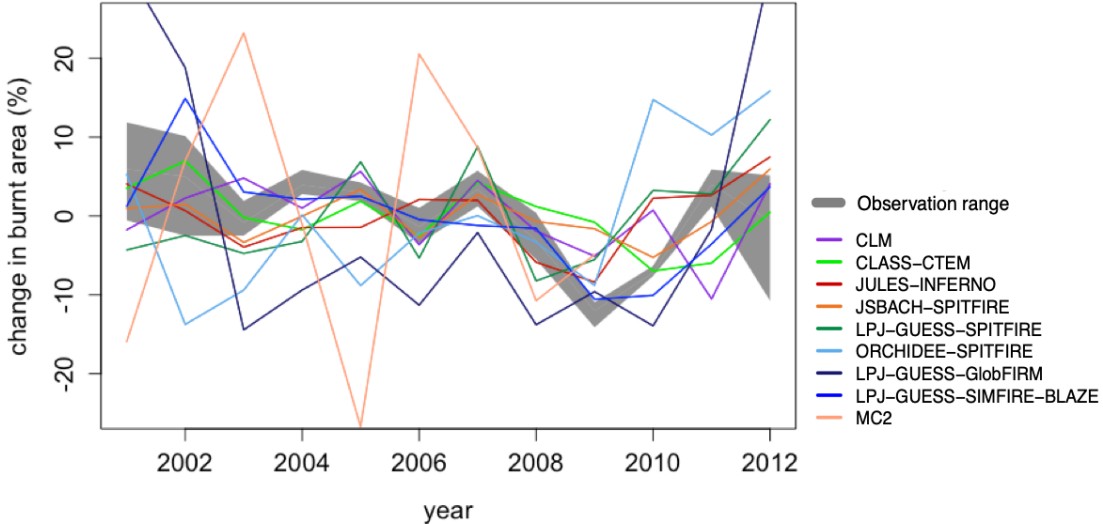

**Figure 4: The range in inter-annual variability in burnt area for the years 2001-2012 for all models and burnt area datasets which span the entire time period (GFED4, GFED4s, MCD45, FireCCI51). Results from the individual**
**FireMIP models, as well as the observational minimum-maximum values, are plotted.**

### 3.3 Overall assessment

Our evaluation suggests that LPJ-GUESS-GlobFIRM and MC2 produce substantially poorer simulations of burnt area and its inter-annual variability than other models in the FireMIP ensemble. These are both older models,

developed before the availability of global burnt area products (in the case of LPJ-GUESS-GlobFIRM) or calibrated regionally and not designed to run at global scale (MC2). While the other models perform better in simulating fire properties, there is no single model that outperforms other models across the full range of fire and vegetation benchmarks examined here. Model structure does not explain the differences in model performance. Process-based fire models (see table 1) appear to be slightly better able to represent the spatial pattern in burnt area

than empirical models (mean score 0.87 and 0.94 respectively), but this difference is largely the result of including

GlobFIRM in the empirical model ensemble; removing this model results in a mean score of 0.87 for these models. The inter-model spread in scores within each group is much larger than the difference between the two types of model. Only one empirical model simulates fire seasonality, but this model performs worse than each of the process-based models, independent of reference dataset considered. There is no difference in the performance of process-based and empirical models with respect to IAV in burnt area, seasonal phase in burnt area or fire emissions.

The FireMIP simulations include three models in which versions of the same process-based fire module (SPITFIRE) are coupled to different vegetation models. These three models produce a wide range of benchmarking scores for burnt area, with mean benchmarking scores of 0.79, 0.85 and 0.72 for JSBACH, LPJ-GUESS and ORCHIDEE respectively. There are also large differences between these models in terms of other aspects of the fire regime (Table 3). As there are only moderate differences between the different SPITFIRE implementations (Rabin et al., 2017), this suggests that the overall difference between the models reflect interactions between the fire and vegetation modules.

Models using prescribed vegetation biogeography (CLM, CLASS-CTEM, JSBACH-SPITFIRE, ORCHIDEE-SPITFIRE) represent the spatial pattern of burnt area better than models with dynamic vegetation (JULES-INFERNO, LPJ-GUESS-SPITFIRE, LPJ-GUESS-GlobFIRM, LPJ-GUESS-SIMFIRE-BLAZE, MC2), with respective mean benchmarking scores across all burnt area data sets of 0.79 and 0.97. This difference is still present even when LPJ-GUESS-GlobFIRM and MC2 are not included (0.90). It seems likely that models using prescribed vegetation biogeography have a better representation of fuel loads and flammability. This can also partially be seen in the positive relationship between the benchmarking scores of vegetation carbon and burnt area spatial patterns for at least the GFED4, FireCCI4.0 and FireCCI5.1 burnt area reference datasets (mean $R^2$ = 0.31, range 0.19-0.38). Areas where the FireMIP models represent vegetation carbon poorly coincide with some of the regions where models have trouble representing the spatial pattern of burnt area such as dryland regions (Figure 1). Although there is no relationship between GPP/NPP and burnt area benchmarking scores, there is a positive relationship between simulated burnt area scores and the seasonal concentration of GPP (R2 = 0.30-0.84) and, to a lesser extent, the seasonal phase of GPP (R2 = 0.09-0.24). Models which correctly predict the seasonal pattern of GPP/NPP, which has a strong influence on the availability of fuel, are more likely to predict the burnt area correctly. This supports the idea that the seasonality of vegetation production and senescence, is among the chief drivers of the interactions between vegetation and fire within each model. However, since fires combust some of the vegetation and hence reduce fuel loads, fire occurrence also influence the seasonality in vegetation productivity. This may partly explain the varying strength of the correlations between seasonal concentration and phase, and burnt area. Although both GPP/NPP and burnt area are affected by climate conditions, the emergent relationships between simulated and observed climate and burnt area are generally similar across the FireMIP models (Forkel et al., 2019), whereas the emergent relationships between vegetation properties and burnt area are much less so. This indicates that fire models could be improved by improving the simulated fuel availability by a better representation of the seasonality of vegetation production and senescence.

Fire carbon emission benchmarking scores are strongly related to the burnt area performance ($R^2 > 0.85$ for GFED4s and MCD45 and >0.45 for FireCCI4.0 and GFED4). This indicates that simulated burnt area is the main driver of fire emissions, overriding spatial patterns in fuel availability and consumption. However, the benchmarking scores for the spatial pattern in burnt area are better overall than those for fire carbon emissions.

Models that explicitly simulate the impact of human suppression on fire growth or burnt area (CLM, CLASS-CTEM, JSBACH-SPITFIRE, LPJ-GUESS-SIMFIRE-BLAZE) are better at representing the spatial pattern in burnt area compared to models which do not include this effect (0.85 and 0.93 respectively). In the case of the three process-based models (CLM, CLASS-CTEM, JSBACH-SPITFIRE) this is most probably because the spatial pattern in fire size is better represented (Table 3).

CLM is the only model that incorporates cropland fires (Table 1) and it is also the only model which captures the spring peak in burnt area in the northern extratropics associated with crop fires (e.g. Le Page et al., 2010; Magi et al., 2012, Hall et al., 2016). This might also contribute to the good overall score of CLM for spatial pattern of burnt area.

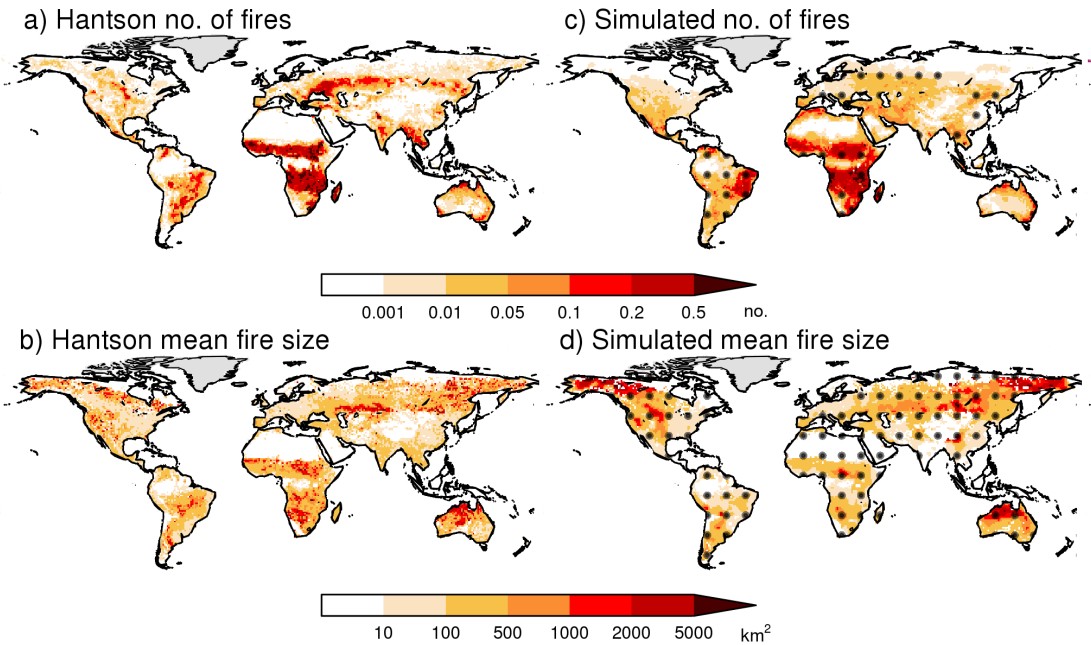

**Figure 5: Reference datasets and mean of three models for number of fires and mean fire size. Model output is adapted so that mean and variance coincide with observations as the total values are not directly comparable (See Supplementary Information S1). Stippling indicates where variance between models is less than the model ensemble mean.**

## 4 Discussion

There are large differences in the total burnt area between the FireMIP models, with two models (LPJ-GUESS-GlobFIRM and MC2) falling well outside the observed range in burnt area for the recent period. In the case of LPJ-GUESS-GlobFIRM, this is because GlobFIRM was developed before global burnt area products were available, resulting in a general poor performance (Kloster and Lasslop, 2017), in combination with the fact that structural changes were made to the vegetation model without a commensurate development of the fire module. In the case of MC2, this probably reflects the fact that MC2 was developed for regional applications but was applied globally here without any refinement of the fire model. The other FireMIP models used the burned area datasets to develop and tune their models. They therefore capture the global spatial patterns of burnt area reasonably well, although no model simulates the very high burnt area in Africa and Australia causing a general underestimation of burnt area in tropical regions and overestimation in extratropical regions. The analysis of a limited number of models suggests that process-based fire models do not simulate the spatial patterns in fire size well (Table 3). In particular they fail to represent fire size in tropical savannas (Figure 5), most probably because they assume a fixed

maximum fire duration of less than one day (Hantson et al., 2016) while savanna fires are often very long-lived (e.g. Andela et al., 2019). Our results suggest that process-based fire models could be improved by a better representation of fire duration. Although none of the FireMIP models simulate multi-day fires, there are fire models that do (e.g. Pfeifer et al., 2013; Le Page et al., 2015) and which could therefore provide a template for future model development. New parameterizations would need to incorporate aspects of natural and anthropogenic landscape fragmentation which limit fire growth (e.g. Pfeifer et al., 2013; Le Page et al., 2015; Kelley et al. 2019). Indeed, our results show that models that include a human limitation on fire growth represent the global spatial pattern in burnt area and fire size better. The recently generated Global Fire Atlas (Andela et al., 2019) includes aspects of the fire behaviour (e.g., fire spread rate and duration), which offer new opportunities to examine and parameterize fire.

Vegetation type and stocks are input variables for the fire models, influencing fire ignition and spread in the process-based models and determining simulated burnt area in the empirical models. The occurrence of fire can, in turn, affect the vegetation type, simulated vegetation productivity (i.e. GPP, NPP) and hence the amount and seasonality of fuel build up. Our results indicate that inter-model differences in burnt area are related to differences in simulated vegetation productivity and carbon stocks. Seasonal fuel build-up and senescence is an important driver of global burnt area. Furthermore, we find that models which are better at representing the seasonality of vegetation production are also better at representing the spatial pattern in burnt area. These results are consistent with the analysis of emergent relationships in FireMIP models, which shows the need to improve processes related to plant production and biomass allocation to improve model performance in simulating burnt area (Forkel et al., 2019a). While there are spatially explicit global estimates regarding carbon stocks in live vegetation, there is limited information about carbon stocks of different fuel types and how these change between seasons and over time (van Leeuwen et al., 2014; Pettinari & Chuvieco, 2016). Furthermore, fuel availability could be substantially affected by livestock density and pasture management (Andela et al., 2017). While improved representation of land management practices could improve the representation of fire, the lack of high-quality fuel availability data currently limits our ability to constrain simulated fuel loads.

The FireMIP models generally do not simulate the timing of peak fire occurrence accurately and tend to simulate a fire season longer than observed. This might be related to the representation of seasonality in vegetation production and fuel build up. However, human activities can also change the timing of fire occurrence (e.g. Le Page et al., 2010; Rabin et al., 2015), and so an improved representation of the human influence on fire occurrence and timing could also help to improve the simulated fire seasonality. The importance of the anthropogenic impact on fire seasonality is especially clear in the northern extratropics (e.g. Archibald et al., 2009; Le Page et al., 2010; Magi et al., 2012), where the only model that explicitly includes crop fires (CLM) is also the only model that shows the bimodal seasonality. Thus, the inclusion of anthropogenic fires could help to improve model simulations. However, this requires a better understanding of how fire is used for land management under different socio-economic and cultural conditions (Pfeiffer et al., 2013; Li et al., 2013).

Global inter-annual variability in burnt area is largely driven by drought episodes in high biomass regions and fuel buildup after periods of increased rainfall in dryland areas (e.g. Chen et al., 2017). Previous analysis has shown that the FireMIP models are relatively good at representing emergent climate-fire relationships (Forkel et al., 2019a); hence it seems plausible that fuel build up and its effect on subsequent burnt area is not well represented in the models and that this is the reason for the poor simulation of IAV in burnt area. This is in line with our

findings and the findings of Forkel et al. (2019a) that fire models are not sensitive enough to previous previous-season vegetation productivity.

The spread in simulated global total fire emissions is even larger than for burnt area, but fire emissions largely follow the same spatial and temporal patterns as burnt area (Figure 1, table 3). However, the benchmark scores for emissions are worse than those for burnt area. This reflects the fact that emissions are the product of both errors in simulated vegetation and burnt area. Furthermore, spatial and temporal uncertainties in the completeness of biomass combustion will affect the emissions. While improvements to vegetation and fuel loads are likely to

produce more reliable estimates of emissions, an improved representation of the drivers of combustion completeness in models will also be required for more accurate fire emission estimates. Only one of the FireMIP models (CLM) includes cropland, peatland, and deforestation fire explicitly, albeit in a rather simple way. Our analyses suggest that this does not produce an improvement in the simulation of either the spatial pattern or timing of carbon emissions. However, given that together these fires represent a substantial proportion of annual carbon

emissions, a focus on developing and testing robust parameterisations for these largely anthropogenic fires could also help to provide more accurate fire emission estimates.

    Our analysis demonstrates that benchmarking scores provide an objective measure of model performance and can be used to identify models that might negatively impact on a multi-model mean and so exclude these from further analysis (e.g. LPJ-GUESS-GlobFIRM, MC2). At the moment, a further ranking is more difficult because

no model clearly outperforms all other models. Still, some FireMIP models are better at representing some aspects of the fire regime compared to others. Hence, when using FireMIP output for future analyses, one could weigh the different models based on the score for the variable of interest, thus giving more weight to models which perform better for these variables.

*Competing interests.* The authors declare no competing interest

       *Author contribution.* SH and DK analysed the model results; SH, DB, MF, GL, FL, SM, JRM provided global fire-vegetation model outputs; SH, DK, SPH and AA wrote the manuscript with input from all authors.

*Code and data availability.* The benchmarking code is archived https://zenodo.org/record/3879161#.Xtq-py-z2fU (https://doi.org/10.5281/zenodo.3879161), which also contains the code to produce the figures presented here. The FireMIP model output is archived at https://zenodo.org/record/3555562#.Xell3C2ZOcY (DOI:10.5281/zenodo.3551041). Data availability for each reference dataset is provided in the table below:

| Variable | Dataset | data access |
| --- | --- | --- |
| Burnt area | GFED4 | https://www.globalfiredata.org/data.html |
| | GFED4S | https://www.globalfiredata.org/data.html |
| | MCD45 | Upon request to the author |
| | FireCCI40 | https://geogra.uah.es/fire_cci/ |
| | FireCCI51 | https://geogra.uah.es/fire_cci/ |
| Fire emissions | GFAS | https://apps.ecmwf.int/datasets/data/cams-gfas/ |
| Fire size & number | Hantson | https://zenodo.org/record/3564818#.Xelidi2ZOcY |
| GPP | Luyssaert | Upon request to the author |
| | Jung | https://www.bgc-jena.mpg.de/geodb/projects/Home.php |
| NPP | Michaletz | https://onlinelibrary.wiley.com/action/downloadSupplement ?doi=10.1111%2Fgeb.12685&file=geb12685-sup-0002-suppinfo2.xlsx |
| | Luyssaert | Upon request to the author |
| | EMDI | http://gaim.unh.edu/Structure/Intercomparison/EMDI/validationdata2/ |
| LAI | MCD15 | http://doi.org/10.5067/MODIS/MCD15A2H.006 |
| | AVHRR | https://www.ncei.noaa.gov/data/avhrr-land-leaf-area-index-and-fapar/access/ |
| Carbon in vegetation | Avitabile | http://lucid.wur.nl/datasets/high-carbon-ecosystems |
| | Carvalhais | Upon request to the author |


       *Acknowledgments.* S. Hantson, S. Rabin, and A. Arneth acknowledge support by the EU FP7 projects BACCHUS (grant agreement no. 603445) and LUC4C (grant ag. No. 603542). This work was supported, in part, by the German
Federal Ministry of Education and Research (BMBF), through the Helmholtz Association and its research programme ATMO, and the HGF Impulse and Networking fund. S. P. Harrison acknowledges support under the ERC-funded project GC2.0 (Global Change 2.0: Unlocking the past for a clearer future, grant number 694481). D. Kelley was supported by the UK Natural Environment Research Council through The UK Earth System Modelling Project (UKESM, Grant No. NE/N017951/1). GL was funded by the Deutsche Forschungsgemeinschaft (DFG,
German Research Foundation – 338130981. F. Li is supported by National Natural Science Foundation of China (41475099 and 41875137). LN acknowledges financial support from the Strategic Research Area MERGE (Modeling the Regional and Global Earth System - www.merge.lu.se), and from the Lund University Centre for Climate and Carbon Cycle Studies (LUCCI).

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

Table 1: Brief description of the global fire models that ran the FireMIP baseline experiments. Process indicates models which explicitly simulate ignitions and fire spread. A detailed overview can be found in Rabin et al. (2017).

| Model | Dynamic Biogeography | Fire model type | Human suppression of fire spread/ burnt area | Spatial resolution (lon x lat) | Temporal resolution | Reference |
|---|---|---|---|---|---|---|
| CLM | No | Process | Yes | 2.5° x 1.9° | Half hourly | Li et al., 2013 |
| CLASS-CTEM | No | Process | Yes | 2.8125° x 2.8125° | Daily | Melton and Arora, 2016 |
| JULES-INFERNO | Yes, but without fire feedback | Empirical | No | 1.875° x 1.245° | Half hourly | Mangeon et al., 2016 |
| JSBACH-SPITFIRE | No | Process | Yes | 1.875° x 1.875° | Daily | Lasslop et al., 2014 |
| LPJ-GUESS-SPITFIRE | Yes | Process | No | 0.5° x 0.5° | Daily | Lehsten et al., 2009 |
| LPJ- GUESS-GlobFIRM | Yes | Empirical | No | 0.5° x 0.5° | Annual | Smith et al., 2014 |
| LPJ-GUESS-SIMFIRE-BLAZE | Yes | Empirical | Yes | 0.5° x 0.5° | Annual | Knorr et al., 2016 |
| MC2 | Yes | Process | No | 0.5° x 0.5° | Monthly | Bachelet et al., 2015 |
| ORCHIDEE-SPITFIRE | No | Process | No | 0.5° x 0.5° | Daily | Yue et al., 2014 |


**Table 2: Simulated and observed burnt area (Mha) for the period 2002-2012 for the globe and for key regions including the northern extratropics (NET, > 30°N), the southern extratropics (SET, > 30°S), the tropics (30°N - 30°S), the savanna regions of Africa (18°W-40°E & 13°N-20°S), the savanna region of South America (42°-68°W & 9°S-25°S), Australian Savanna (120°E-155°E & 11°S-20°S), the agricultural band of central Eurasia (30°E-85°E & 50°N-58°N), the Mediterranean basin (10°W-37°E & 31°N-44°N), and the western USA (100°-125°W & 31°N-43°N). Data availability for FireCCI40 is limited to 2005-2011 and for MC2 to 2002-2009.**


| | Global | NET | Tropics | SET | S-American savanna | African savanna | Australian savanna | Central Eurasia | Mediterranean basin | western USA |
|---|---|---|---|---|---|---|---|---|---|---|
| GFED4s | 468 | 39 | 426 | 4 | 18 | 295 | 35 | 8.5 | 1.3 | 1.0 |
| GFED4 | 349 | 27 | 319 | 3 | 14 | 218 | 34 | 5.2 | 0.8 | 0.9 |
| MCD45 | 348 | 33 | 312 | 4 | 13 | 232 | 25 | 7.0 | 2.0 | 0.9 |
| FireCCI40 | 345 | 23 | 320 | 2 | 8 | 237 | 25 | 6.8 | 1.1 | 0.8 |
| FireCCI51 | 387 | 37 | 347 | 3 | 14 | 230 | 38 | 10.2 | 1.3 | 1.1 |
| CLM | 454 | 77 | 362 | 15 | 36 | 194 | 15 | 7.9 | 9.3 | 3.4 |
| CLASS-CTEM | 536 | 41 | 466 | 28 | 46 | 172 | 20 | 2.0 | 4.3 | 9.5 |
| JULES-INFERNO | 381 | 76 | 292 | 13 | 26 | 128 | 23 | 5.0 | 11.0 | 7.7 |
| JSBACH-SPITFIRE | 457 | 114 | 318 | 25 | 21 | 166 | 17 | 15.5 | 9.5 | 9.7 |
| LPJ-GUESS-GlobFIRM | 39 | 14 | 24 | 1 | 3 | 7 | 3 | 0.6 | 0.6 | 0.5 |
| LPJ-GUESS-SPITFIRE | 393 | 99 | 280 | 14 | 51 | 135 | 2.8 | 12.5 | 14.5 | 6.1 |
| LPJ-GUESS-SIMFIRE-BLAZE | 482 | 86 | 381 | 15 | 72 | 146 | 27 | 3.4 | 7.9 | 14.9 |
| MC2 | 97 | 40 | 54 | 3 | 2 | 17 | 2 | 0.9 | 5.0 | 2.2 |
| ORCHIDEE-SPITFIRE | 471 | 16 | 435 | 19 | 13 | 246 | 81 | 2.4 | 2.4 | 0.3 |


**Table 3: Benchmarking scores after removing the influence of differences in the mean and variance for each individual global fire model for key fire and vegetation variables. A lower score is "better", with a perfect score equal to 0. The full table with all benchmarking scores is presented in Table S1. Dataset information can be found in Supplementary Information S1. LPJ-G = LPJ-GUESS. Cell are coloured blue if the benchmarking score is lower than both null models, yellow if lower than 1 null model and red when higher than both null models.**


| | Dataset | Mean | random | CLM | CLASS-CTEM | JULES INFERNO | JSBACH SPITFIRE | LPJ-G GlobFIRM | LPJ-G SPITFIRE | SIMFIRE-BLAZE | MC2 | ORCHIDEE SPITFIRE |
|---|---|---|---|---|---|---|---|---|---|---|---|---|
| **Burnt area** | | | | | | | | | | | | |
| spatial | GFED4s | 1 | 1.07 | 0.63 | 0.79 | 0.72 | 0.70 | 1.06 | 0.94 | 0.88 | 1.00 | 0.72 |
| | GFED4 | 1 | 1.14 | 0.80 | 0.93 | 0.85 | 0.86 | 1.08 | 0.98 | 0.88 | 1.07 | 0.71 |
| | MCD45 | 1 | 1.16 | 0.65 | 0.81 | 0.72 | 0.69 | 1.12 | 0.93 | 0.92 | 1.02 | 0.70 |
| | FireCCI40 | 1 | 1.13 | 0.77 | 0.98 | 0.89 | 0.92 | 1.09 | 0.93 | 0.97 | 1.13 | 0.73 |
| | FireCCI51 | 1 | 1.11 | 0.83 | 1.01 | 0.91 | 0.93 | 1.11 | 0.96 | 0.97 | 1.23 | 0.70 |
| seasonal phase | GFED4s | 0.56 | 0.22 | 0.12 | 0.12 | 0.13 | 0.12 | | 0.31 | | | 0.31 |
| | GFED4 | 0.49 | 0.47 | 0.34 | 0.35 | 0.41 | 0.42 | | 0.33 | | | 0.31 |
| | MCD45 | 0.56 | 0.26 | 0.12 | 0.11 | 0.12 | 0.12 | | 0.30 | | | 0.30 |
| | FireCCI40 | 0.60 | 0.12 | 0.16 | 0.43 | 0.17 | 0.16 | | 0.33 | | | 0.32 |
| | FireCCI51 | 0.55 | 0.25 | 0.26 | 0.28 | 0.33 | 0.32 | | 0.32 | | | 0.31 |
| seasonal concentration | GFED4s | 1 | 1.36 | 1.16 | 1.15 | 1.24 | 1.15 | | 1.13 | | | 1.22 |
| | GFED4 | 1 | 1.35 | 1.19 | 1.12 | 1.25 | 1.11 | | 1.18 | | | 1.19 |
| | MCD45 | 1 | 1.36 | 1.14 | 1.08 | 1.26 | 1.13 | | 1.12 | | | 1.20 |
| | FireCCI40 | 1 | 1.34 | 1.31 | 1.26 | 1.36 | 1.25 | | 1.29 | | | 1.30 |
| | FireCCI51 | 1 | 1.36 | 1.25 | 1.22 | 1.33 | 1.21 | | 1.20 | | | 1.27 |
| IAV | GFED4s | 1 | 1.46 | 1.17 | 0.65 | 1.18 | 1.09 | 0.66 | 1.36 | 0.76 | 1.66 | 1.44 |
| | GFED4 | 1 | 1.27 | 0.98 | 1.62 | 1.23 | 0.89 | 1.04 | 1.08 | 1.00 | 1.41 | 1.25 |
| | MCD45 | 1 | 1.32 | 0.93 | 1.34 | 1.11 | 0.84 | 0.73 | 0.97 | 1.27 | 1.67 | 1.22 |
| | FireCCI5.1 | 1 | 1.42 | 1.18 | 1.53 | 1.24 | 1.27 | 1.73 | 1.27 | 1.23 | 1.87 | 1.12 |
| **fire emission** | | | | | | | | | | | | |
| spatial | GFAS | 1 | 1.08 | 0.78 | 0.85 | 0.73 | 0.74 | 1.13 | 1.03 | 0.91 | 1.06 | 0.86 |
| seasonal phase | GFAS | 0.78 | 0.18 | 0.16 | 0.20 | 0.17 | 0.15 | | 0.37 | | | 0.34 |
| seasonal concentration | GFAS | 1 | 1.36 | 1.20 | 1.22 | 1.30 | 1.17 | | 1.27 | | | 1.25 |
| IAV | GFAS | 1 | 1.36 | 0.77 | 1.70 | 1.28 | 1.09 | 1.42 | 1.42 | 1.11 | 1.41 | 1.49 |
| **Fire number** | | | | | | | | | | | | |
| spatial | Hantson | 1 | 1.19 | | | | 0.96 | | 0.83 | | | 0.76 |
| **Fire size** | | | | | | | | | | | | |
| Spatial | Hantson | 1 | 1.31 | | | | 1.03 | | 1.22 | | | 1.12 |
| **GPP** | | | | | | | | | | | | |
| spatial | Luyssaert | 1 | 1.39 | 1.49 | 1.41 | 1.46 | 1.39 | 1.41 | 1.24 | 1.37 | | 1.09 |
| spatial | Jung | 1 | 1.30 | 0.64 | 0.46 | 0.39 | 0.42 | 0.46 | 0.67 | 0.43 | | 0.49 |
| seasonal phase | Jung | 0.42 | 0.65 | 0.18 | 0.23 | 0.19 | 0.23 | | 0.22 | | | 0.22 |
| seasonal concentration | Jung | 1 | 1.65 | 1.08 | 1.19 | 1.14 | 1.21 | | 1.19 | | | 1.09 |
| **NPP** | | | | | | | | | | | | |
| spatial | Michaletz | 1 | 1.39 | 0.82 | 0.79 | 0.77 | 0.75 | 0.96 | 0.86 | 0.89 | 0.88 | 0.99 |
| spatial | Luyssaert | 1 | 1.33 | 0.90 | 1.01 | 0.53 | 0.76 | 0.82 | 0.87 | 0.79 | 0.68 | 0.84 |
| spatial | EMDI | 1 | 1.30 | 0.91 | 0.87 | 0.58 | 0.66 | 0.79 | 0.83 | 0.81 | 0.65 | 0.80 |
| **LAI** | | | | | | | | | | | | |
| spatial | MCD15 | 1 | 1.29 | 0.60 | 0.53 | 0.44 | 0.78 | 0.70 | 0.61 | 0.57 | 0.63 | |
| spatial | AVHRR | 1 | 1.29 | 0.81 | 0.71 | 0.49 | 0.65 | 0.74 | 0.62 | 0.61 | 0.64 | |
| **Carbon in vegetation** | | | | | | | | | | | | |
| spatial | Avitabile | 1 | 1.32 | 0.69 | 0.88 | 0.76 | 0.78 | 0.76 | 0.76 | 0.74 | 0.80 | 0.70 |
| spatial | Carvalhais | 1 | 1.32 | 0.66 | 0.66 | 0.58 | 0.64 | 0.62 | 0.66 | 0.58 | 0.67 | 0.54 |