# Peer review of "Quantitative assessment of fire and vegetation properties in simulations with fire-enabled vegetation models from the Fire Model Intercomparison Project"

_Geoscientific Model Development, 2019_

## Referee Comment (RC1) · Anonymous Referee #1 · 5 Feb 2020

The authors provide a detailed glimpse into the successes and struggles of global fire modeling efforts, and quantitatively try to isolate the most pressing challenges for both individual fire models and the fire modeling community as a whole by using a benchmarking method of comparisons with observations. Particularly interesting is that the authors highlight how sensitive the benchmarking results are to how vegetation (fuel load) is captured or simulated in any particular fire model. I think the paper should be published after a few minor revisions and/or author responses/clarifications to my concerns below.

Comments

Title: this is a confusing title because nowhere in the paper are the historical FireMIP simulation results discussed. Line 145 and essentially all the figures point out that only present day results are analysed. "Historical" in the CMIP framework usually refers to simulation periods that extend from about 1850 to present day. I would strongly suggest changing the title to better capture the scope of the analysis the authors undertook.

Lines 95-104: this paragraph is difficult to follow relative to the analysis of FireMIP output. Are the authors trying to say that benchmarking allows for a more systematic evaluation of models so that a hierarchy can be quantifiably justified? If so, I suggest that the authors add or clarify this in the text to make it clear to readers that this is why the authors chose to raise this discussion point. Alternatively, the authors could shorten or delete the paragraph altogether, because while they raise the point of ending model democracy, it stands in contrast with the conclusions of the study, where the authors say the "no model clearly outperforms all other models" which seems to be avoiding the issue of hierarchical treatment of the fire models. If this group of authors cannot ascribe a hierarchy to global fire models, then I think they miss the chance to advance the conversation from the perspective of their collective expertise. By this, I mean that I, as the reader, can walk away from the paper with useful benchmarks and metrics, but that I will also then evaluate model quality on my own because the authors did not. My conclusion is that while the benchmarks are great to have, the results in Figure 2 and Table 2 clearly show that GlobFIRM and MC2 output should not be considered equally alongside output from other models.

Paragraph at line 166: Certainly there are observational uncertainties, but the Global Fire Atlas and other studies about fire products (GFED papers and MODIS papers, at least) have made a solid effort to quantify uncertainties – what do the authors suggest is enough in terms of validation of the observations? Some specific problems I have with the paragraph: In line 169, saying "large uncertainties still remain for most variables" is too vague. Which variables? How large, or large compared with what? To me,

it seems that fire models have larger uncertainty than the observations. I would argue that the results in this paper suggest that model uncertainty does not arise from a lack of observations, but rather, the model uncertainty is largely due to poor simulations of biomass. While this paragraph makes it sound like models are waiting for observations of bulk properties, it is more accurate to say that the fire models do not have the fuel process simulated correctly. These are two different issues that should not be about a lack of observational constraints. I suggest the paragraph be shortened a sentence or two so that the focus of the paper remains on evaluation of model output, and not observations. The authors could simply point out that burnt area, biomass, and fire emissions estimates vary and uncertainty is still being characterized, and cite appropriate papers. To me, this paper is about the benchmarking results, and the fact that observations have weaknesses too should be relegated to a side note with citations.

Table 3: Why are the benchmarking scores for the Mean null model often equal to 1? Is this an artifact of the calculation itself? If so, wouldn't this detract from the utility of using the Mean null model as a point of comparison with fire model benchmark scores for those fire variables?

Paragraph at line 229: The text discussion seems inconsistent with the results in Table 3. I may be misunderstanding the reason for the benchmarking scores for the Mean and Random null models, but my interpretation is that those Mean and Random null model benchmarking scores are the target to beat. If a fire model beats that benchmark score, then my interpretation is that that particular fire model performs better than the null model. Is that a correct interpretation? If so, then there seem to be some inconsistencies between the text and Table 3 as follows.

Paragraph at line 229: Specifically, one sentence states "The models capture the timing of the peak fire season reasonably well, with all of the models performing better than both null models for seasonal phase in burnt area" but many of the fire models have benchmark scores greater than the Random null model, so why do the authors say "all"?

Paragraph at line 229: Another sentence states "all of the FireMIP models perform worse than both null models for seasonal concentration of burnt area, independent of the reference burnt area dataset" but looking at Table 3, almost all of the fire model benchmark scores are less than the benchmark scores for the Random null model, with the exception being JULES-INFERNO vs FireCCI40. Wouldn't this mean that the comparisons are all better than the Random null model?

Future model development section: I would suggest that the authors propose mechanisms that fire models should include (crops, prescribed biogeography), and reflect on both why some fire models do not include those mechanisms already, and whether the future of fire model development will include those mechanisms. Or perhaps this is discussed in other FireMIP papers already? Also, the authors might provide a broader perspective in this section by discussing whether there are global fire models currently in use that did not participate in FireMIP but do include features that the benchmarking results in this study highlight as particularly weak. For example, Pfeiffer et al's LPJ-LMFire model https://www.geosci-model-dev.net/6/643/2013/ includes representation of human use of fire in a novel way.

---

## Referee Comment (RC2) · Sergey Venevsky (Referee) · 25 Mar 2020

The paper presents evaluation of historical simulations made for the nine FireMIP models with regard to fire and vegetation properties. This is very important and necessary inter comparison study aimed to support and move further global fire modelling activities. Both methods and results are clearly presented and scientifically sounded and proven. The only methodological weakness is a bit short period of comparison of the models with the observations which could be clearly longer. I think that Discussion in this paper is the weakest part and needs more effort to make conclusions from the

model inter-comparison to be stronger and more clear. In particularly 1. The part about relation of areas burnt to vegetation production (lines 387 -396 and 405-411) should describe in more details why and how exactly fuel load influence burnt areas in the FIREmip nine models 2. Part on influence of areas burnt upon GPP/NPP is absent and should be added

Comments/questions/suggestions: Line 66 – "Willdfires and anthropogenic fires" – how you define and classify these types of fires in global models? Further on no clarifications for this important question... Line 126 Lightning data 1900-1920.population density and land use 1700 where do these data come from? Line 132 The baseline FireMIP simulation is a transient experiment starting in 1700 CE and continuing to 2013. Why simulation is only up to 2013 and comparison is only for 2002-2012 ? Can simulation be somehow extended to include recent years? Similarly, inter-comparison only for decade looks not so sounded, for example 1997/1998 El Ninio years are out... Which climate data was used? Lines 150-159 What was the principle of selection of all these datasets? Why no global water cycle related datasets (e.g. runoff) where selected? Water status is obviously important for both fire and vegetation, I would compare at least also runoff for 2002-2012 Line 173 "As model benchmarking techniques become more sophisticated it would be beneficial to better evaluate the datasets the models are compared against to ensure the models are being benchmarked appropriately" Please, delete or rephrase (shorten) Line 175-180. I think you should move formulae of NME from Supplementary back to the main text. You write in Supplementary that you applied NME for areas burnt, but it is clear from Table 3 and S1 that you apply the same metrics for other variables for benchmarking, please, correct. As well Table 3 is quite difficult to read, why not to use semaphore colors (not so good- red, OK –yellow, good –green) or any other color scheme? Figure 1p. Performance in fire emissions by the models is the worst from all variables. How you can explain it? How reliable is observation data set? Please, make more explanation in Discussion part. Line 219 CLM (NME: 0.63-0.80) and ORCHIDEE-SPITFIRE (0.70-0.73) are the best performing models. What makes these models to be the best in burnt areas description (for CLM

as I understood it is related to cropland fires, what about ORCHIDEE-SPITFIRE ? Line 326 to 334 "... the overall difference between the models (...JSBACH, LPJ-GUESS and ORCHIDEE..) reflect feedbacks between the fire and vegetation modules " what are these feedbacks? Where lays difference in their descriptions of these three models (in DGVMs)? Lines 345-349 "there is a positive relationship between simulated burnt area scores and the seasonal concentration of GPP (R2 = 0.30-0.84) and, to a lesser extent, the seasonal phase of GPP (R2 = 0.09-0.24). This supports the idea that seasonal vegetation production and senescence, which have an important influence on fuel loads, drive the interactions between vegetation and fire within each model" – I doubt this statement. It is more likely that similar dynamics of burnt areas and the seasonal concentration of GPP/ the seasonal phase of GPP are related to dependence of both areas burnt and GPP variables from soil moisture in the fire models and DGVMs. Please, either prove, or delete this paragraph.

I wish all the best for FIREmip in their further valuable research activities

Sergey Venevsky

―――――――――――――――――――――

---

## Author Comment (AC1) · 6 May 2020

**Author response to Anonymous Referee #1 review**

*The authors thank referee #1 for this considered comments and constructive suggestions.*

*Below we provide a detailed response in italic to each comment.*

The authors provide a detailed glimpse into the successes and struggles of global fire modeling efforts, and quantitatively try to isolate the most pressing challenges for both individual fire models and the fire modeling community as a whole by using a benchmarking method of comparisons with observations. Particularly interesting is that the authors highlight how sensitive the benchmarking results are to how vegetation (fuel load) is captured or simulated in any particular fire model. I think the paper should be published after a few minor revisions and/or author responses/clarifications to my concerns below.

Comments

Title: this is a confusing title because nowhere in the paper are the historical FireMIP simulation results discussed. Line 145 and essentially all the figures point out that only present day results are analysed. "Historical" in the CMIP framework usually refers to simulation periods that extend from about 1850 to present day. I would strongly suggest changing the title to better capture the scope of the analysis the authors undertook.

*The simulations we examined are indeed historical, in the sense that they were run from 1700 CE to the present day, although we only evaluate them in the recent past because of the availability of data. But we agree the title might imply evaluation over a longer period, and we will change it to: "Quantitative assessment of fire and vegetation properties in simulations with fire-enabled vegetation models from the Fire Model Intercomparison Project".*

Lines 95-104: this paragraph is difficult to follow relative to the analysis of FireMIP output. Are the authors trying to say that benchmarking allows for a more systematic evaluation of models so that a hierarchy can be quantifiably justified? If so, I suggest that the authors add or clarify this in the text to make it clear to readers that this is why the authors chose to raise this discussion point. Alternatively, the authors could shorten or delete the paragraph altogether, because while they raise the point of ending model democracy, it stands in contrast with the conclusions of the study, where the authors say the "no model clearly outperforms all other models" which seems to be avoiding the issue of hierarchical treatment of the fire models. If this group of authors cannot ascribe a hierarchy to global fire models, then I think they miss the chance to advance the conversation from the perspective of their collective expertise. By this, I mean that I, as the reader, can walk away from the paper with useful benchmarks and metrics, but that I will also then evaluate model quality on my own because the authors did not. My conclusion is that while the benchmarks are great to have, the results in Figure 2 and Table 2 clearly show that GlobFIRM and MC2 output should not be considered equally alongside output from other models.

*We agree that the GlobFIRM and MC2 simulations are poor and not comparable to the other simulations in the FireMIP ensemble, and indeed we state this (lines 369-370). The reviewer indeed interprets the objective of this paragraph correctly as we think that establishing a hierarchy of model's ability to simulate fire is important (and hence we would like to keep the paragraph explaining that this is one of the goals of benchmarking) and we should have made a stronger statement about this in the abstract and conclusion. We will modify the text as follows:*

- *Line 57 et seq. "The two older fire models included in the FireMIP ensemble (LPJ-GUESS-GlobFIRM, MC2) clearly perform less well globally than other models, but it is difficult to distinguish between the remaining ensemble members: some of these models are better at representing certain aspects of the fire regime, none clearly outperforms all other models across the full range of variables assessed."*
- *Line 319: "Our evaluation suggests that LPJ-GUESS-GlobFIRM and MC2 produce substantially poorer simulations of burnt area and its inter-annual variability than other models in the FireMIP ensemble. These are both older models, developed before the availability of global burnt area products (in the case of LPJ-GUESS-GlobFIRM) or calibrated regionally and not designed to run at global scale (MC2). While the other models perform better in simulating fire properties, there is no single model that outperforms other models across the full range of fire and vegetation benchmarks examined here. Model structure does not explain the differences in model performance."*
- *We furthermore included an extra paragraph at the end of the discussion to cover this point: "Our analysis demonstrates that benchmarking scores provide an objective measure of model performance and can be used to identify models that might negatively impact on a multi-model mean and so exclude these from further analysis (e.g. LPJ-GUESS-GlobFIRM, MC2). At the moment, a further ranking is more difficult because no model clearly outperforms all other models. Still, some FireMIP models are better at representing some aspects of the fire regime compared to others. Hence, when using FireMIP output for future analyses, one could weigh the different models based on the score for the variable of interest, thus giving more weight to models which perform better for these variables."*

Paragraph at line 166: Certainly there are observational uncertainties, but the Global Fire Atlas and other studies about fire products (GFED papers and MODIS papers, at least) have made a solid effort to quantify uncertainties – what do the authors suggest is enough in terms of validation of the observations? Some specific problems I have with the paragraph: In line 169, saying "large uncertainties still remain for most variables" is too vague. Which variables? How large, or large compared with what? To me, it seems that fire models have larger uncertainty than the observations. I would argue that the results in this paper suggest that model uncertainty does not arise from a lack of observations, but rather, the model uncertainty is largely due to poor simulations of biomass. While this paragraph makes it sound like models are waiting for observations of bulk properties, it is more accurate to say that the fire models do not have the fuel process simulated correctly. These are two different issues that should not be about a lack of observational constraints. I suggest the paragraph be shortened a sentence or two so that the focus of the paper remains on evaluation of model output, and not observations. The authors could simply point out that burnt area, biomass, and fire emissions estimates vary and uncertainty is still being characterized, and cite appropriate papers. To me, this paper is about the benchmarking results, and the fact that observations have weaknesses too should be relegated to a side note with citations.

*We agree that the focus of this paper should be on the evaluation of the model results. Our intention in this paragraph was definitely not to critique the groups producing different fire datasets or to imply that they are not trying to provide both theoretical (Brennan et al., 2019) and practical uncertainty estimates (e.g. Giglio et al., 2013), but to explain why we do not take account* of observational uncertainties in our comparisons. *We agree with the reviewer that the uncertainty in model output exceeds the uncertainty of existing datasets and we agree that this paragraph might distract, and we will shorten it drastically and reduced it to its essence, rewriting it as follows:*

*"Ideally, model benchmarking should take account of uncertainties in the observations, for example by down-weighting less reliable data sets (e.g. Collier et al. 2018). However, observational uncertainties are not reported for some of the data sets used here (e.g. vegetation carbon). Furthermore, some of the data sets (e.g. emissions) involve modelled relationships; there has been little formal assessment of the choice of model on the resultant observational uncertainty. While we use multiple datasets when available (e.g. for burnt area, where there are large differences between the products), in an attempt to integrate observational uncertainty in our evaluations, it seems premature to incorporate uncertainty in the benchmark data sets in a formal sense when calculating the benchmarking scores."*

Table 3: Why are the benchmarking scores for the Mean null model often equal to 1? Is this an artifact of the calculation itself? If so, wouldn't this detract from the utility of using the Mean null model as a point of comparison with fire model benchmark scores for those fire variables?

*The normalized mean error (NME) is constructed in such a way as to normalize the scores against an objective background so that the mean null model results in a score = 1 (Kelley et al., 2013, Biogeosciences 10: 3313-3340). This is not an artifact but a design feature of the metric to make the interpretation of the results more intuitive compared to other error metrics. All the values shown as less than 1 in Table 3 are for seasonal phase and are calculated using the mean Phase Difference metric, which is not constrained in the same way. Since our description of the metrics is not clear, and also in response to comments by the second reviewer, we have rewritten the section of text describing the metrics and the null models as follows:*

*"To assess model ability to reproduce spatial patterns in a variable, we use the normalised mean error (NME):*

$$NME = \frac{\sum A_i |obs_i - sim_i|}{\sum A_i |obs_i - \overline{obs}|}$$  (1)

*where the difference between observations (obs) and simulation (sim) are summed over all cells (i) weighted by cell area ($A_i$) and normalized by the average distance from the mean of the observations ($\overline{obs}$). Since NME is proportional to mean absolute errors, the smaller the NME value the better the model performance. A score of 0 represents a perfect match to observations. NME has no upper bound.*

*NME can be sensitive to the simulated magnitude of the variable. To take this into account in comparisons, we removed the influence of biases in the mean and variance between model results and each reference dataset. This has the further desirable property of limiting the impact of observational uncertainties in the reference datasets on the comparisons. Although we focus on benchmarking results after removing biases in the mean and variance, the scores for comparisons before this procedure (and for comparisons after removing mean biases only) are given in Supplementary Information S2.*

*To assess model ability to reproduce seasonal patterns in a variable, we focused on seasonal concentration (roughly equivalent to the inverse of season length) and seasonal phase (or timing). We calculated a mean seasonal "vector" for each observed and simulated location based on the monthly distribution of the variable through the year. The concentration is the length of this vector compared to the annual value, and ranges between 0 when the variable is distributed evenly throughout the year and 1 when the season is confined to a single month. The phase is indicated by the direction of the vector. Observed and modelled concentrations were compared using NME. Phase is compared using the Mean Phase Difference (MPD) metric (see Supplementary Information S2). Again, for NME, a score of 0 represents a perfect match to observations and*

*there is no upper bound. MPD has a maximum value of 1 when all cells have a maximum phase mismatch of 6 months. Seasonality metrics could not be calculated for three models (LPJ-GUESS-GlobFIRM, LPJ-GUESS-SIMFIRE-BLAZE, MC2), either because they do not simulate the seasonal cycle or because they did not provide these outputs. We did not use FireCC4.0 to assess seasonality or interannual variability (IAV) in burnt area because it has a much shorter times series than the other burnt area products.*

*Model scores are interpreted by comparing them to two null models (Kelley et al., 2013). The "mean" null model compares each benchmark dataset to a dataset of the same size created using the mean value of all the observations. The mean null model for NME always has a value of 1 because the metric is normalised by the mean difference. The mean null model for MPD is based on the mean direction across all observations, and therefore the value can vary and is always less than 1. The "randomly-resampled" null model compares the benchmark data set to these observations resampled 1000 times without replacement (Table 3). The "randomly-resampled" null model is normally worse than the mean null model for NME comparisons. For MPD, the mean will be better than the random null model when most grid cells show the same phase. A detailed description of the benchmarking metrics is given in the Supplementary Information S2. "*

Paragraph at line 229: The text discussion seems inconsistent with the results in Table 3. I may be misunderstanding the reason for the benchmarking scores for the Mean and Random null models, but my interpretation is that those Mean and Random null model benchmarking scores are the target to beat. If a fire model beats that benchmark score, then my interpretation is that that particular fire model performs better than the null model. Is that a correct interpretation? If so, then there seem to be some inconsistencies between the text and Table 3 as follows.

*Your interpretation is correct. We stated this in the original manuscript (lines 192-198) but we have now rewritten the section on the metrics and their interpretation (as described above) and hope this is now clearer. The confusions between the Table and the text are due to mistakes on our part in the description and we have now corrected these, as explained below.*

Paragraph at line 229: Specifically, one sentence states "The models capture the timing of the peak fire season reasonably well, with all of the models performing better than both null models for seasonal phase in burnt area" but many of the fire models have benchmark scores greater than the Random null model, so why do the authors say "all"?

*This sentence should have read "The models capture the timing of the peak fire season reasonably well, with all of the models performing better than the mean null model for seasonal phase in burnt area". And have added additionally: "The models also frequently perform better than the random null model, with all models performing better against GFED4.".*

Paragraph at line 229: Another sentence states "all of the FireMIP models perform worse than both null models for seasonal concentration of burnt area, independent of the reference burnt area dataset" but looking at Table 3, almost all of the fire model benchmark scores are less than the benchmark scores for the Random null model, with the exception being JULES-INFERNO vs FireCCI40. Wouldn't this mean that the comparisons are all better than the Random null model?

*This should have read "mean null model" instead of "both null models" and has been corrected.*

Future model development section: I would suggest that the authors propose mechanisms that fire models should include (crops, prescribed biogeography), and reflect on both why some fire models do not include those mechanisms already, and whether the future of fire model development will include those mechanisms. Or perhaps this is discussed in other FireMIP papers already? Also, the authors might provide a broader perspective in this section by discussing whether there are global fire models currently in use that did not participate in FireMIP but do include features that the benchmarking results in this study highlight as particularly weak. For example, Pfeiffer et al's LPJ-LMFire model https://www.geosci-model-dev.net/6/643/2013/ includes representation of human use of fire in a novel way.

*We agree that the title of this section is somewhat misleading, since we do not believe it is possible, as yet, to prescribe exactly the steps that would yield an improved fire model. Our intention here was to point to areas which need to be investigated further because the benchmarking identifies them as weaknesses in the current models. It would be possible, for example, to include crops into the models or human use of fire (as in LPJ-LMFire). However, the current parameterizations of agricultural fires are relatively simple and generally not based on rigorous data analysis. And indeed, as the ongoing discussion about the impacts of anthropogenic activity on fire trends shows, our understanding of human-fire interactions is very incomplete. Similarly, we have identified the ability to reproduce vegetation properties and hence fuel loads as an area where the models do not perform well -- but again, this is an active area of research and, as yet, there is no agreed way forward. However, we agree that it would be valuable to point out which processes are already implemented in different fire models not participating in FireMIP (including LPJ-LMFire) and will adapt the discussion section in different points accordingly. We will also re-title this section simply as: Discussion.*

*References*

*Brennan, J., Gómez-Dans, J. L., Disney, M., and Lewis, P.: Theoretical uncertainties for global satellite-derived burned area estimates, Biogeosciences, 16, 3147-3164, 10.5194/bg-16-3147-2019, 2019.*
*Hall, J. V., Loboda, T. V., Giglio, L., and McCarty, G. W.: A MODIS-based burned area assessment for Russian croplands: Mapping requirements and challenges, Remote sensing of environment, 184, 506-521, http://dx.doi.org/10.1016/j.rse.2016.07.022, 2016.*
*Roteta, E., Bastarrika, A., Padilla, M., Storm, T., and Chuvieco, E.: Development of a Sentinel-2 burned area algorithm: Generation of a small fire database for sub-Saharan Africa, Remote Sensing of Environment, 222, 1-17, https://doi.org/10.1016/j.rse.2018.12.011, 2019.*

---

## Author Comment (AC2) · 6 May 2020

**Author response to Sergey Venevsky's review**

*The authors thank Sergey Venevsky for his constructive comments. Below we provide a detailed respond in italics to each comment and suggestion.*

The paper presents evaluation of historical simulations made for the nine FireMIP models with regard to fire and vegetation properties. This is very important and necessary inter comparison study aimed to support and move further global fire modelling activities. Both methods and results are clearly presented and scientifically sounded and proven. The only methodological weakness is a bit short period of comparison of the models with the observations which could be clearly longer. I think that Discussion in this paper is the weakest part and needs more effort to make conclusions from the model intercomparison to be stronger and more clear. In particularly 1. The part about relation of areas burnt to vegetation production (lines 387 -396 and 405-411) should describe in more details why and how exactly fuel load influence burnt areas in the FIREmip nine models 2. Part on influence of areas burnt upon GPP/NPP is absent and should be added.

*The global fire models participating in FireMIP use output from the vegetation models as input (e.g. different fuel loads, fAPAR, etc.) to a range of fire processes, depending on the fire model parameterization and structure. A detailed description of the fire model structure, and he interactions with the vegetation model is outside the scope of this paper and has been described in Hantson et al. (2016) and Rabin et al. (2017). However, we acknowledge the need to indicate that this interaction is present and will now modify the text to make this clearer (line 387):*
*"Vegetation type and stocks are input variables for the fire models, influencing fire ignition and spread in the process-based models and determining simulated burnt area in the empirical models. The occurrence of fire can, in turn, affect the vegetation type, simulated vegetation productivity (i.e. GPP, NPP) and hence the amount and seasonality of fuel build up. Our results indicate that inter-model differences in burnt area are related to differences in simulated vegetation productivity and carbon stocks. Seasonal fuel build-up and senescence is an important driver of global burnt area. Furthermore, we find that models which are better at representing the seasonality of vegetation production are also better at representing the spatial pattern in burnt area."*

Comments/questions/suggestions:

Line 66 – "Willdfires and anthropogenic fires" – how you define and classify these types of fires in global models? Further on no clarifications for this important question. . .

*We make the distinction here between lightning-ignited fires and fires set by humans. Some models include human ignitions as a function of population density; one model (CLM) also includes cropland fires. The specific parameterizations included in the FireMIP models are discussed in the FireMIP protocol papers (Hantson et al., 2016; Rabin et al., 2017). Since we are not documenting the models here, but only evaluating their performance we do not think that a full discussion of these parameterizations is warranted. However, we will modify this sentence to clarify this point:*
*"However, the representation of both lightning-ignited fires and anthropogenic fires (including cropland fires) varies greatly in global fire models."*

Line 126 Lightning data 1900-1920.population density and land use 1700 where do these data come from?

*The source of the data sets is given in the FireMIP protocol, which we cite here (line 133). However, we will modify this to make it clear that the information about all the drivers of the simulation are available there, as follows: "(see Rabin et al., 2017 for description of the modelling protocol and the sources of the input data for the experiments).".*

Line 132 The baseline FireMIP simulation is a transient experiment starting in 1700 CE and continuing to 2013. Why simulation is only up to 2013 and comparison is only for 2002-2012? Can simulation be somehow extended to include recent years? Similarly, inter-comparison only for decade looks not so sounded, for example 1997/1998 El Ninio years are out. . . Which climate data was used?

*The simulation protocol was drawn up in 2016 and at that stage there was no readily available driving data post-2013. The choice of the interval 2002-2012 for benchmarking was motivated by two considerations. First, 2002 was the first year when both MODIS sensors were operative and hence a temporally coherent high quality global burnt area dataset is available. Before 2002, and especially before 2000, the quality of the GFED4 archive is much lower. Second, two modelling groups did not run the final year of the simulation (i.e. 2013). We therefore excluded this year in order to keep the evaluation between the models as consistent as possible. CRU-NCEP climate was used as forcing data for the simulations (see Rabin et al., 2017).*
*We agree that there are several other evaluations of fire models that could be made and certainly testing how well they emulate the response to El Nino variability would be an interesting case study. However, this is out of scope for the current round of FireMIP.*

Lines 150-159 What was the principle of selection of all these datasets? Why no global water cycle related datasets (e.g. runoff) where selected? Water status is obviously important for both fire and vegetation, I would compare at least also runoff for 2002-2012

*We agree that there are many other opportunities for evaluating fire models, and it would be possible to include data sets such as runoff. However, there are a number of issues that led us not to include runoff in our evaluation: (1) runoff data in the most fire-prone regions (e.g. the Sahel) are not very reliable; (2) gauged runoff is integrated property, and thus requires models to incorporate some form of hydrological routing scheme; and (3) it is not the most relevant hydrological variable for fire -- soil moisture or litter moisture would be useful but are very heterogenous and thus global evaluation would be difficult. We agree that we could usefully modify this sentence to explain our current focus and will modify the text to read:*
*"Model performance was evaluated using site-based and remotely sensed global data sets of fire occurrence, fire-related emissions and vegetation properties (Figure 1; Figure S1). We include vegetation variables (e.g. GPP, NPP, biomass, LAI) because previous analyses have indicated that they are critical for simulating fire occurrence and behaviour (Forkel et al., 2019a; Teckentrup et al.,2019) and there are global data sets available. We did not consider parameters such as soil or litter moisture because, although these may have an important influence on fire behaviour, globally comprehensive data sets are not available."*

Line 173 "As model benchmarking techniques become more sophisticated it would be beneficial to better evaluate the datasets the models are compared against to ensure the models are being benchmarked appropriately" Please, delete or rephrase (shorten)

*We have substantially shortened and modified this paragraph based on your feedback as well as the comments by the first reviewer (see response to reviewer 1 for detailed information regarding the changes made).*

Line 175-180. I think you should move formulae of NME from Supplementary back to the main text. You write in Supplementary that you applied NME for areas burnt, but it is clear from Table 3 and S1 that you apply the same metrics for other variables for benchmarking, please, correct.

*Agreed. We have put the formula of NME to the main manuscript. Additionally, we have substantially rewritten the section on the benchmarking metrics to make the procedure clearer, including clarifying that NME is used for all the spatial variables.*

As well Table 3 is quite difficult to read, why not to use semaphore colors (not so good- red, OK –yellow, good –green) or any other color scheme?

*We thank the reviewer for his suggestion. As the objective of this table is to present the numerical scores so that readers will be able to judge differences between the models based on the actual scores, we keep these in the table as well. However, we have now added background colors to identify large-scale differences between the scores. We will modify the caption to this table as follow: "Cell are coloured blue if the benchmarking score is lower than both null models, yellow if lower than 1 null model and red when higher than both null models.".*

Figure 1p. Performance in fire emissions by the models is the worst from all variables. How you can explain it? How reliable is observation data set? Please, make more explanation in Discussion part.

*We discuss why the scores for emissions are worse than the scores for other aspects of fire already in the final paragraph of the Discussion. As we state there, errors in the emissions are a product of errors in burnt area, simulated biomass and combustion completeness. It is therefore natural that they are more difficult to predict accurately.*

Line 219 CLM (NME: 0.63-0.80) and ORCHIDEE-SPITFIRE (0.70-0.73) are the best performing models. What makes these models to be the best in burnt areas description (for CLM as I understood it is related to cropland fires, what about ORCHIDEE-SPITFIRE ?

*We focus on model structure and processes in this manuscript, and only focus on an individual model if it is the only one representing a certain process (e.g. CLM and cropland fires). In this case, both models prescribe vegetation cover, and we have shown that vegetation is one of the variables influencing performance in simulated burnt area. In fact, both models have the best scores at simulated vegetation carbon.*

Line 326 to 334 ". . . the overall difference between the models (. . .JSBACH, LPJ-GUESS and ORCHIDEE..) reflect feedbacks between the fire and vegetation modules " what are these feedbacks? Where lays difference in their descriptions of these three models (in DGVMs)?

*Fires in DGVMs combust biomass & grass and kill trees, hence exerting a strong impact on vegetation dynamics, which themselves drive fire occurrence and characteristics. The three models mentioned are completely different vegetation models. While fire can impact a large range of processes in LPJ-GUESS, ranging from carbon stocks, over distribution and structure. This is less the case for JSBACH and ORCHIDEE as these do not represent vegetation structure and*

*prescribe vegetation distribution. We provide a very detailed description of each fire model in the FireMIP protocol paper (Rabin et al., 2017) and we also provide the key reference providing the description of each vegetation model. We feel that the description of the structure of each vegetation model is outside the scope of this manuscript.*

Lines 345-349 "there is a positive relationship between simulated burnt area scores and the seasonal concentration of GPP (R2 = 0.30-0.84) and, to a lesser extent, the seasonal phase of GPP (R2 = 0.09-0.24). This supports the idea that seasonal vegetation production and senescence, which have an important influence on fuel loads, drive the interactions between vegetation and fire within each model" – I doubt this statement. It is more likely that similar dynamics of burnt areas and the seasonal concentration of GPP/ the seasonal phase of GPP are related to dependence of both areas burnt and GPP variables from soil moisture in the fire models and DGVMs. Please, either prove, or delete this paragraph.

*We agree that soil moisture, vegetation productivity and fire occurrence are strongly linked to each other and it could be difficult to separate their individual influences. Some recent fire models explicitly represent fuel moisture (e.g. SPITFIRE; Thonicke et al., 2010), with the objective of diagnosing the role of fuel moisture on fire spread. However, soil moisture is only one of the multiple variables which drive fuel moisture, which results in a partial disconnection between soil moisture and fuel moisture in the models. Furthermore, the influence of differences in soil moisture dynamics between models is likely to be small in our experiments because the climate inputs controlling this (precipitation, temperature) were specified to be the same. Thus, we doubt that this feature is solely induced by soil moisture. Furthermore, in addition to this relationship between seasonal vegetation production and burnt area, we provide multiple other indicators that vegetation status impacts the performance of the fire module. This was also a conclusion from a previous study on FireMIP outputs (Forkel et al., 2019). Hence, we believe that there is enough evidence to support our statement that our results stress the importance of the interactions between vegetation and fire within each model.*

*References*

*Forkel, M., Andela, N., Harrison, S. P., Lasslop, G., van Marle, M., Chuvieco, E., Dorigo, W., Forrest, M., Hantson, S., Heil, A., Li, F., Melton, J., Sitch, S., Yue, C., and Arneth, A.: Emergent relationships with respect to burned area in global satellite observations and fire-enabled vegetation models, Biogeosciences, 16, 57-76, 10.5194/bg-16-57-2019, 2019.*

*Hantson, S., Arneth, A., Harrison, S. P., Kelley, D. I., Prentice, I. C., Rabin, S. S., Archibald, S., Mouillot, F., Arnold, S. R., Artaxo, P., Bachelet, D., Ciais, P., Forrest, M., Friedlingstein, P., Hickler, T., Kaplan, J. O., Kloster, S., Knorr, W., Lasslop, G., Li, F., Mangeon, S., Melton, J. R., Meyn, A., Sitch, S., Spessa, A., van der Werf, G. R., Voulgarakis, A., and Yue, C.: The status and challenge of global fire modelling, Biogeosciences, 13, 3359-3375, 10.5194/bg-13-3359-2016, 2016.*

*Rabin, S. S., Melton, J. R., Lasslop, G., Bachelet, D., Forrest, M., Hantson, S., Kaplan, J. O., Li, F., Mangeon, S., Ward, D. S., Yue, C., Arora, V. K., Hickler, T., Kloster, S., Knorr, W., Nieradzik, L., Spessa, A., Folberth, G. A., Sheehan, T., Voulgarakis, A., Kelley, D. I., Prentice, I. C., Sitch, S., Harrison, S., and Arneth, A.: The Fire Modeling Intercomparison*

*Project (FireMIP), phase 1: experimental and analytical protocols with detailed model descriptions, Geosci. Model Dev., 10, 1175-1197, 10.5194/gmd-10-1175-2017, 2017.*

*Thonicke, K., Spessa, A., Prentice, I. C., Harrison, S. P., Dong, L., and Carmona- Moreno, C.: The influence of vegetation, fire spread and fire behaviour on biomass burning and trace gas emissions: results from a process-based model, Biogeosciences, 7, 1991-2011, 10.5194/bg-7-1991-2010, 2010.*

---

## Author Response (AR2)

**We thank Dr. Hargreaves for the detailed comments and suggestions. Below we give a detailed response in *italic* below each comment.**

Comments to the Author:

5  1) The metrics used in this evaluation paper are defined in the supplement and their description does not include reference to previous work. This leads me to think that these metrics have been developed here specifically for evaluating these models. Even if this is not the case, GMD has a general geoscientific audience who will not necessarily be aware of the norms in the field. The definitions, plus full context and reasoning for the choice of metrics should appear in the main text of the

10  manuscript, not in the supplement. Likewise for the description of the datasets used - please describe these fully in the main manuscript. The extra figures and tables can probably remain in the supplement.

On the details of the metrics, for example, I know there are arguments on both sides, but for a model-data comparison which surely aims towards consideration of how models may be improved,

15  RMSE not MAE would be my natural choice. (See for example, doi:10.5194/gmd-7-1247-2014, "in the data assimilation field, the sum of squared errors is often defined as the cost function to be minimized by adjusting model parameters. In such applications, penalizing large errors through the defined least-square terms proves to be very effective in improving model performance.")

*The benchmarking metrics that we are using are those defined in "Kelley et al. (2013). A comprehensive benchmarking system for evaluating dynamic global vegetation models. Biogeosciences Discussions 9: 1-63. Biogeosciences 10: 3313-3340."*

25  *This provides a standard for the evaluation of vegetation models, including fire-enabled vegetation models. We do in fact cite this paper when we define the null models (line 257). However, we apologise that we did not make it clear in the general description of our approach that we were adopting this protocol for benchmarking. We clarify that we are using the Kelley et al. protocol as follow:*

*We will modify the last paragraph of the Introduction as follows:*

*"In this paper, we focus on quantitative evaluation of model performance using the baseline historical simulation and a range of vegetation and fire observational datasets. We use the vegetation-model evaluation framework described by Kelley et al. (2013), with an extended set of data targets to quantify the fire and vegetation properties and their uncertainties. We identify (i) common weaknesses of the*

35  *current generation of global fire-vegetation models, (ii) factors causing differences between the models, and (iii) discuss the implications for future model development."*

*We will also clarify our approach in the Method section, including the choice of NME, by adding a paragraph at the beginning of 2.2 as follows:*

40

*"We adopted the metrics and comparison approach specified by Kelley et al. (2013) as it provides a comprehensive scheme for the evaluation of vegetation models. This protocol provides specifically designed metrics to quantify model performance in terms of annual average, seasonal and inter-annual variability against a range of global data sets, allowing the impact of spatial and temporal biases in*

45  *means and variability to be assessed separately. The derived model scores were compared to scores based on the temporal or spatial mean value of the observations and a "random" model produced by bootstrap resampling of the observations.*

*NME was selected over other metrics (e.g. RMSE) as these normalised scores allow for direct comparison in performance between variables with different units (Kelley et al., 2013), NME is more*

50  *appropriate for variables which do not follow a normal distribution and it has therefore been used as the standard metric to asses global fire model performance (e.g, Kloster & Lasslop, 2017; kelley et al., 2019, Boer et al., 2020) . NME is defined as:"*

*We will also include the citation to Kelley et al. (2013) when we define NME and MPD to make it clear*

55  *that these metrics are derived from that paper.*

*We furthermore moved the full description of the metrics to the methods section. While the target data sets are all standard, we are happy to move the description of these also into the main text (although we will leave the figures in Supplementary). To do so we have restructured the methods section by slitting*

60 *section 2.2 into a new section 2.2 "Benchmarking reference datasets" and moving most of the previous section 2.2. to section 2.3 "Model Evaluation and Benchmarking". We also moved the definition of the metrics used from the supplementary material to the methods section. We reworked the text so that the relevant information from the supplementary material (both the information on the metrics and the datasets used) fits in neatly.*

65

2) Although it may be premature to include data uncertainties in the numerical metrics, more discussion of these uncertainties is required. In particular, the point that reviewer 1 makes about comparing the variation within the ensemble with the uncertainty in the data is valid. This is

70 something it is important to have a feel for. Again, remember that some of your readers will be generally interested in model-data evaluation, but innocent of the details of fire-enabled models and the relevant data.

*Very few of the targets unfortunately allow to provide formal uncertainty estimates. Uncertainty can*

75 *therefore almost only be assessed by comparison with other data sets. This is why we use multiple data sets in our comparisons. The choice of partitioning method for GPP estimates has been shown to have an impact, but the upscaling algorithms have not been (to our knowledge) formally tested. By moving the description of the data sets into the main text, the uncertainties involved in the production of individual targets should be clearer to the readers. However, we will expand the paragraph describing*

80 *why we do not integrate uncertainty into calculation of the metrics as follows:*

*"Ideally, model benchmarking should take account of uncertainties in the observations. However, observational uncertainties are not reported for most of the data sets used here (e.g. vegetation carbon). While it would in principle be possible to include uncertainty for example by down-weighting less*

85 *reliable data sets (e.g. Collier et al. 2018), determining the merits of the methods used to obtain observational data is rather subjective and no agreement as to which is more reliable if multiple reference datasets exist for the same variable (e.g. burnt area). Furthermore, some of the data sets (e.g. emissions) involve modelled relationships; there has been little assessment of the impact of the choice of model on the resultant uncertainty in emission estimates (e.g. Kaiser et al., 2012). While we use*

90 *multiple datasets when available (e.g. for burnt area, where there are extremely large differences between the products and they may all underestimate the actual burnt area (Roteta et al., 2019)), in an attempt to integrate observational uncertainty in our evaluations, it seems premature to incorporate uncertainty in the benchmark data sets in a formal sense in calculating the benchmarking scores."*

95

3) Section 3.1
"The simulated modern (2002-2012) total global annual burnt area is between 39 and 536 Mha (Table 2). Most of the FireMIP models are within the range of the remotely sensed observed burnt area (354 to 468 Mha a−1)."

100 This does not seem to me to be a fair description of the ensemble. Table 2 suggests that two models are very far from the observed range but the rest are impressively close to the quoted range, although I may be misunderstanding the data uncertainty (which I am assuming is 354-468 at 1 standard deviation).

105 *The data uncertainty here is actually the range between different burnt area products, it is not the uncertainty at one standard deviation. Most burnt area products do not provide uncertainty estimates and the use of multiple products is the only way we have to make an assessment of uncertainty (see e.g. Forkel et al., 2019). Even this may be underestimating the real uncertainty because several of the data sets rely on the same active fire product from MODIS. Furthermore, recent work using Sentinel 2*

110 *products (unfortunately not yet available globally) suggests that all of the products may underestimate burnt area. We will modify the text to make it clearer that the quoted range is not the statistical uncertainty but rather that estimate (without uncertainty) produced by different products. We will change this text to read:*

115 *"The simulated modern (2002-2012) total global annual burnt area is between 39 and 536 Mha (Table 2). Most of the FireMIP models are within the range of burnt area estimated by the individual remotely sensed products (354 to 468 Mha a−1). LPJ-GUESS-GlobFIRM and MC2 simulate much less burnt area*

*than the shown by any of the products and CLASS-CTEM simulates more than shown by any of the products. However, use of the range of the remotely sensed estimates may not be a sufficient measure of the uncertainty in burnt area because four of them are derived from the same active fire product (Forkel et al., 2019) and recent work suggests that they may all underestimate burnt area (Roteta et al., 2019). Thus, we cannot definitively say that the apparent overestimation by CLASS-CTEM is unrealistic."*

4) Reviewer 2 wrote:
"Lines 345-349 "there is a positive relationship between simulated burnt area scores and the seasonal concentration of GPP (R2 = 0.30-0.84) and, to a lesser extent, the seasonal phase of GPP (R2 = 0.09-0.24). This supports the idea that seasonal vegetation production and senescence, which have an important influence on fuel loads, drive the interactions between vegetation and fire within each model" – I doubt this statement. It is more likely that similar dynamics of burnt areas and the seasonal concentration of GPP/ the seasonal phase of GPP are related to dependence of both areas burnt and GPP variables from soil moisture in the fire models and DGVMs. Please, either prove, or delete this paragraph. "

I don't understand your response to this, as I do not think the reviewer is, as you assert, suggesting that "this feature is solely induced by soil moisture"...? Sorry if I am being hopelessly stupid, but I remain concerned that if this reviewer cannot follow your justification for the existence of this positive relationship, then other readers will also have the same problem. I would have thought that some extra explanation at least should be added to the manuscript even if you cannot go as far as to totally "prove" it.

*We will expand the argument to make it clear why we believe that fuel build up is the key issue explaining why there is a relationship between seasonality in GPP/NPP and burnt area but no relationship between total GPP/NPP and burnt area in section 3.3 as follows:*

*"Although there is no relationship between GPP/NPP and burnt area benchmarking scores, there is a positive relationship between simulated burnt area scores and the seasonal concentration of GPP ($R^2 = 0.30$-$0.84$) and, to a lesser extent, the seasonal phase of GPP ($R^2 = 0.09$-$0.24$). Models which correctly predict the seasonal pattern of GPP/NPP, which has a strong influence on the availability of fuel, are more likely to predict the burnt area correctly. This supports the idea that the seasonality of vegetation production and senescence, is among the chief drivers of the interactions between vegetation and fire within each model. However, since fires combust some of the vegetation and hence reduce fuel loads, fire occurrence also influence the seasonality in vegetation productivity. This may partly explain the varying strength of the correlations between seasonal concentration and phase, and burnt area. Although both GPP/NPP and burnt area are affected by climate conditions, the emergent relationships between simulated and observed climate and burnt area are generally similar across the FireMIP models (Forkel et al., 2019), whereas the emergent relationships between vegetation properties and burnt area are much less so. This indicates that fire models could be improved by improving the simulated fuel availability by a better representation of the seasonality of vegetation production and senescence."*

5) Thank you for making the benchmarking code available and including a nice readme file. It is not clear to me whether turnkey code for the creation of the figures has also been included. If not, please add this. If it is already there, please flag it clearly in your readme file and also in the Code and Data availability section.

*The code for most figures was already included within the benchmarking code. We now added the code for the missing figures and indicated in both the readme file and the Code and Data availability section that the code to produce the figures is also provided there.*

**List of relevant changes made in the manuscript**

175
1) We adapted the last paragraph of the introduction to indicate the benchmarking scheme of Kelley et al. (2013) was used.
2) We reworked the methods section, moving all information regarding benchmarking metrics and reference datasets from the supplementary here. an extra section on Benchmarking reference datasets has been added to improve the methods structure.
180
3) We adapted the first paragraph of the results section to make observational data range clearer.
4) We reworked a paragraph in the results section to regarding to improve our explanation on the importance of vegetation production and senescence on modelled fire occurrence.
5) We added information to code availability to clarify that the code to produce the figures can be found together with the benchmarking code.

[revised manuscript text omitted]